# SCBench: A KV Cache-Centric Analysis of Long-Context Methods

**Yucheng Li[†], Huiqiang Jiang[‡], Qianhui Wu, Xufang Luo, Surin Ahn, Chengruidong Zhang,
Amir H. Abdi, Dongsheng Li, Jianfeng Gao, Yuqing Yang, Lili Qiu**
Microsoft Corporation, [†]University of Surrey
yucheng.li@surrey.ac.uk,{hjiang,yuqyang}@microsoft.com
https://aka.ms/SCBench

## Abstract

Long-context Large Language Models (LLMs) have enabled numerous downstream applications but also introduced significant challenges related to computational and memory efficiency. To address these challenges, optimizations for long-context inference have been developed, centered around the KV cache. However, existing benchmarks often evaluate in single-request, neglecting the full lifecycle of the KV cache in real-world use. This oversight is particularly critical, as KV cache reuse has become widely adopted in LLMs inference frameworks, such as vLLM and SGLang, as well as by LLM providers, including OpenAI, Microsoft, Google, and Anthropic. To address this gap, we introduce **SCBench** (**S**hared**C**ontext**BENCH**), a comprehensive benchmark for evaluating long-context methods from a KV cache-centric perspective: 1) *KV cache generation*, 2) *KV cache compression*, 3) *KV cache retrieval*, and 4) *KV cache loading*. Specifically, SCBench uses test examples with shared context, ranging 12 tasks with two shared context modes, covering four categories of long-context capabilities: *string retrieval*, *semantic retrieval*, *global information*, and *multi-task*. With SCBench, we provide an extensive KV cache-centric analysis of eight categories long-context solutions, including Gated Linear RNNs (Codestal-Mamba), Mamba-Attention hybrids (Jamba-1.5-Mini), and efficient methods such as sparse attention, KV cache dropping, quantization, retrieval, loading, and prompt compression. The evaluation is conducted on six Transformer-based long-context LLMs: Llama-3.1-8B/70B, Qwen2.5-72B/32B, Llama-3-8B-262K, and GLM-4-9B. Our findings show that sub-$O(n)$ memory methods suffer in multi-turn scenarios, while sparse encoding with $O(n)$ memory and sub-$O(n^2)$ pre-filling computation perform robustly. Dynamic sparsity yields more expressive KV caches than static patterns, and layer-level sparsity in hybrid architectures reduces memory usage with strong performance. Additionally, we identify attention distribution shift issues in long-generation scenarios.

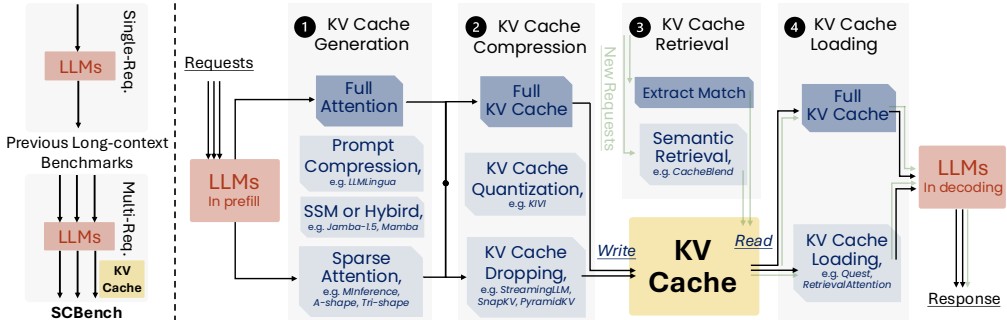

Figure 1: KV Cache lifecycle. Prior benchmarks focus on single-request, while real-world applications reuse KV cache across requests. We propose **SCBench** and categorize long-context methods into KV Cache Generation, Compression, Retrieval, and Loading from a KV-cache-centric perspective.

---

[†]Work during internship at Microsoft. [‡]Corresponding author.

# 1 INTRODUCTION

Long-context capability is becoming a standard for Large Language Models (LLMs), with many of them supporting context windows ranging from 128K to 10M tokens (Reid et al., 2024; Lieber et al., 2024; Dubey et al., 2024; Gradient, 2024). These extended context windows unlock a wide range of real-world applications, such as repository-level code understanding and debugging (Bairi et al., 2024; Park et al., 2023; Liu et al., 2024c; Jimenez et al., 2024), long-document question-answering (Caciularu et al., 2023; Li et al., 2024b), many-shot in-context learning (Agarwal et al., 2024), and long-generation Chain-of-Thought (CoT) reasoning (OpenAI, 2024a; Snell et al., 2024).

Despite the benefit, Long-context inputs also present unique challenges for LLM inference due to high computational costs and memory demands. This has led to the development of efficient long-context solutions leveraging sparsity in various stage of KV cache. In this paper, we introduce an unified analysis framework for efficient long-context methods in a KV cache centric perspective, consisting of the four essential stages of KV cache: 1) KV cache generation, 2) KV cache compression, 3) KV cache retrieval, and 4) KV cache loading. First, **KV cache generation**, aka prefilling, processes the input prompt and produce the KV cache to be used in decoding. In this stage, sparse attention methods are proposed to reduce the complexity of the attention operation (Child et al., 2019; Beltagy et al., 2020; Jiang et al., 2024). Second, **KV cache compression** techniques prune KV states to reduce the memory costs in decoding (Xiao et al., 2024; Li et al., 2024c). Third, **KV cache retrieval** aims to skip KV cache generation of an incoming request, and instead retrieves and reuses KV cache from history KV cache pool for more efficient inference (Zheng et al., 2024; Yao et al., 2024a). At last, **KV cache loading** aims to load only partial of the KV cache for each decoding step to save the memory and computing cost (Tang et al., 2024; Liu et al., 2024b).

However, these methods are only evaluated on single-request benchmarks (Hsieh et al., 2024; Zhang et al., 2024a; Kamradt, 2023; Li et al., 2024a), which fail to covers the full lifecycle of KV cache in real applications. Typically, real-world applications often require reusing prompt memory (i.e., KV cache) and involving multiple requests or multi-round interactions (Qin et al., 2025). The reuse of KV cache, known as prefix caching, is already a crucial component in popular inference frameworks (Zheng et al., 2024; vLLM, 2024) and used by LLM providers (Gemini, 2024; Claude, 2024; OpenAI, 2024b; Azure, 2024). In addition, testing with multiple requests is especially crucial for the long-context methods mentioned earlier, as many achieve efficiency through query-conditioned compression. For instance, Arora et al. (2024) reports that Mamba's compression of previous information based on the current query can prevent it from answering follow-up queries.

To address this gap, we introduce *SCBench*, a benchmark designed to evaluate efficient long-context methods that covers the full lifecycle of KV cache in real-world scenarios, particularly for shared context and multi-round interactions where KV Cache is reused for follow-up queries. As shown in Fig. 2b, SCBench assesses four key long-context abilities across 12 tasks with two shared context modes. Each test example includes a shared context and multiple follow-up queries. The four long-context capabilities and their corresponding tasks are:

1. **String Retrieval Capability:** A fundamental requirement for long-context LLMs is retrieving relevant context with exact matches from long inputs. We extend previous retrieval tasks like NIAH and Multi-NIAH (Kamradt, 2023; Hsieh et al., 2024) by introducing three comprehensive

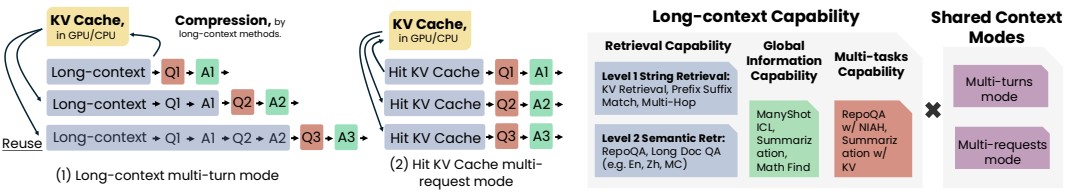

|     (a) Two Shared Context Modes     |     (b) Overview of SCBench     |

Figure 2: Long-context tasks often involve contexts sharing, e.g., multi-turn dialogues, multi-step reasoning, and repository-level tasks. (a) Illustration of two common shared-context patterns. (b) Overview of tasks and scenarios covered by our benchmark, encompassing four categories of long-context abilities and two shared-context modes.

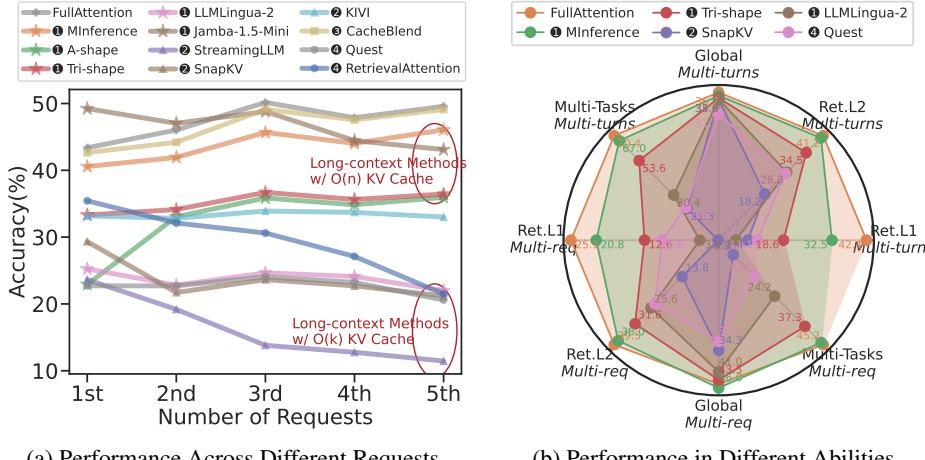

(a) Performance Across Different Requests   (b) Performance in Different Abilities

Figure 3: Overview of performance results for SCBench. (a) Performance trends of various long-context methods across multiple requests. Methods with $O(n)$ memory cost in decoding show improving performance as requests increase. In contrast, methods with sub-$O(n)$ KV cache in decoding, like KV cache dropping methods, perform well only in the first request. (b) Specific performance of different long-context methods across various long-context capability tasks. All evaluated long-context methods exhibit some loss in Retrieval capability while largely maintaining Global Information processing capability.

string retrieval tasks: *key-value* retrieval, *prefix-suffix* retrieval, and *multi-hop* retrieval, measuring capability at different levels of granularity.

2. **Semantic Retrieval Capability:** Real-world applications often require long-context LLMs to understand semantic meaning before succeeding in retrieval. We considered various semantic retrieval scenarios across different domains, building four distinct tests: RepoQA (Liu et al., 2024c) and long-form QA (covering English, Chinese, and multiple-choice questions) (Zhang et al., 2024a).

3. **Global Information Capability:** We also assess the capability of long-context LLMs to process and aggregate global information through three tasks: many-shot in-context learning (Agarwal et al., 2024), summarization, and long array statistics (Zhang et al., 2024a).

4. **Multi-tasking Capability:** In real applications, LLMs often handle multiple tasks with a shared long-context input. Our benchmark evaluates this capability through two tasks: RepoQA with NIAH and summarization with KV retrieval.

In addition, as shown in Fig. 2a, our benchmark includes two typical shared context modes: ***Multi-turn Mode***, where the context is cached within a single session, and ***Multi-request Mode***, where it is cached across multiple sessions.

With SCBench, we conduct an extensive KV cache-centric analysis in the four stages as shown in Fig. 1 (details in §2). Specifically, we evaluate 13 long-context methods across four stages and eight categories on eight open-source long-context LLMs, including Llama-3.1-8B/70B (Dubey et al., 2024), Qwen2.5-72B/32B (Team, 2024), Llama-3-8B-262K (Gradient, 2024), GLM-4-9B-1M (GLM et al., 2024),

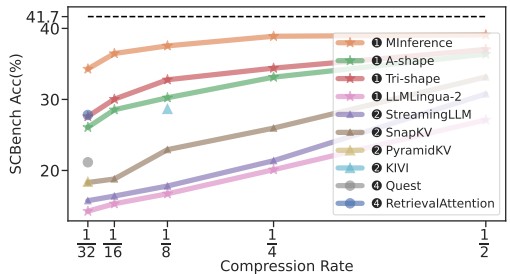

Figure 4: Performance of various long-context methods at different compression rates on SCBench using Llama-3.1-8B (Dubey et al., 2024).

Codestal Mamba (team, 2024), and Jamba-1.5-mini (Lieber et al., 2024). These methods span gated linear RNNs (e.g., Codestal Mamba), hybrid models (e.g., Jamba-1.5), sparse attention (e.g., A-shape, Tri-shape, MInference (Jiang et al., 2024)), prompt compression (e.g., LLMLingua-2 (Pan et al., 2024)), KV cache dropping (e.g., StreamingLLM (Xiao et al., 2024), SnapKV (Li et al., 2024c)), KV cache quantitation (e.g., KIVI (Liu

et al., 2024e)), semantic retrieval (e.g., CacheBlend (Yao et al., 2024a)), and KV cache loading (e.g., Quest (Tang et al., 2024), RetrievalAttention (Liu et al., 2024b)), as detailed in Table 1. Additionally, we introduce Tri-shape, a novel, training-free sparse attention method that demonstrates improved first-turn performance in our evaluations.

Our experimental results reveal the following insights: 1) **Sub-$O(n)$ memory is *almost* infeasible in multi-turn decoding**, as shown in Fig. 3. Sparse decoding methods (sub-$O(n)$ memory) perform well on the first query but lose accuracy in subsequent requests. In contrast, sparse encoding methods ($O(n)$ memory with $O(n^2)$ computation during pre-filling) can approximate full attention accuracy across multiple queries. 2) **Task performance declines at varying rates**, as illustrated in Fig. 3b. Sparse KV cache methods excel in tasks requiring global information, while $O(n)$ memory is crucial for exact match retrieval tasks. 3) **All long-context methods degrade as the budget decreases**, as shown in Fig. 4. However, sub-$O(n)$ memory methods experience a sharp performance drop at a 1/4 compression rate. Methods like RetrievalAttention and KIVI, which maintain $O(n)$ memory with sparse decoding, sustain higher performance even under higher compression rates. 4) **Long-generation scenarios exhibit distribution shift issues**. As generation length and the number of rounds increase, the KV cache's importance distribution changes significantly. This out-of-distribution (OOD) issue leads to performance degradation, even for $O(n)$ memory methods like RetrievalAttention, as shown in Fig. 3.

Our contributions are as follows:

- We propose a new benchmark, SCBench, to evaluate long-context methods on multi-round and multi-request scenarios in two typical KV cache reuse scenarios, providing a more realistic assessment.
- We design an extensive set of downstream tasks, covering four long-context capabilities across 12 subtasks in various domains.
- We systematically categorize long-context methods from a KV-cache-centric perspective and evaluate 13 different long-context methods (including our newly proposed sparse attention method, Tri-shape) on eight state-of-the-art open-source long-context LLMs using SCBench. Our comprehensive analysis reveals key insights into the effects of sparsity in encoding and decoding, task complexity, and more.

## 2 A KV CACHE-CENTRIC PERSPECTIVE ON LONG-CONTEXT METHODS

Table 1: We evaluated long-context methods on SCBench, where $n$ represents the token size of the input prompt and $m$ represents the generation token size, with $n \gg m$.

| Methods | Taxonomy | Stage | P-stage Efficient | D-stage Efficient | KV Cache Size | Prefilling Complexity | Decoding Complexity |
|---|---|---|---|---|---|---|---|
| Codestral Mamba (team, 2024) | Gated Linear RNN | ❶ | ✓ | ✓ | $O(k)$ | $O(kn)$ | $O(km)$ |
| Jamba (Lieber et al., 2024) | Gated Linear RNN + Full Attention | ❶ | ✓ | ✓ | $O(n)$ | $O(n^2)$ | $O(nm)$ |
| LLMLingua-2 (Pan et al., 2024) | Prompt Compression | ❶ | ✓ | ✗ | $O(\alpha n)$ | $O(\alpha^2 n^2)$ | $O(\alpha nm)$ |
| A-shape (Xiao et al., 2024) Tri-shape MInference (Jiang et al., 2024) | Sparse Attention | ❶ | ✓ | ✗ | $O(n)$ | $O(kn)$ | $O(nm)$ |
| StreamingLLM (Xiao et al., 2024) SnapKV (Li et al., 2024c) PyramidKV (Cai et al., 2024) | KV Cache Dropping | ❷ | ✗ | ✓ | $O(k)$ | $O(n^2)$ | $O(km)$ |
| KIVI (Liu et al., 2024e) | KV Cache Quantitation | ❷ | ✗ | ✓ | $O(n)$ | $O(n^2)$ | $O(nm)$ |
| CacheBlend (Yao et al., 2024a) | KV Cache Retrieval | ❸ | ✓ | ✗ | $O(n)$ | $O(n^2)$ | $O(nm)$ |
| Quest (Tang et al., 2024) RetrievalAttention (Liu et al., 2024b) | KV Cache Loading | ❹ | ✗ | ✓ | $O(n)$ | $O(n^2)$ | $O(km)$ |

Recently, a series of works (Gu & Dao, 2024; Xiao et al., 2024; Jiang et al., 2024) have explored various strategies to reduce the inference cost of long-context LLMs, enabling their application (Jimenez et al., 2024; OpenAI, 2024a) to downstream tasks at a lower computational expense. In long-context

LLM inference, the KV cache plays a pivotal role by effectively reducing the computational overhead during the decoding phase. This importance has led to the development of numerous system-level optimizations (Sheng et al., 2023; Qin et al., 2025) focused on KV cache management and scheduling.

In this work, we propose a novel perspective: *these long-context methods can be viewed as optimizations centered around the KV cache at different stages.* Specifically, we introduce a KV-cache-centric framework that systematically categorizes long-context methods into four stages: KV Cache Generation, Compression, Retrieval, and Loading, as illustrated in Fig. 1.

Specifically, the four stages of the KV-cache-centric framework are defined as follows:

1. ***KV Cache Generation:*** This stage optimizes the efficient generation of KV cache during inference. Techniques include sparse attention (e.g., A-shape, Tri-shape, MInference (Jiang et al., 2024), NSA (Yuan et al., 2025), MoBA (Lu et al., 2025)), SSM or hybrid approaches (e.g., Mamba (Gu & Dao, 2024; Dao & Gu, 2024; Lieber et al., 2024)), and prompt compression (e.g., LLMLingua-2 (Pan et al., 2024)).

2. ***KV Cache Compression:*** After generation, the KV cache is compressed before being stored. Methods include KV cache dropping (e.g., StreamingLLM (Xiao et al., 2024), SnapKV (Li et al., 2024c)) and KV cache quantization (e.g., KIVI (Liu et al., 2024e)).

3. ***KV Cache Retrieval:*** Relevant KV cache blocks are retrieved from a storage pool based on the request's prefix, reducing time-to-first-token (TTFT). Approaches include semantic retrieval methods like CacheBlend (Yao et al., 2024a).

4. ***KV Cache Loading:*** This stage dynamically loads the KV cache and computes sparse attention, from KV cache storage (e.g., VRAM, DRAM, SSD, or RDMA) to GPU on-chip SRAM, including Quest (Tang et al., 2024), RetrievalAttention (Liu et al., 2024b), and MagicPIG (Chen et al., 2025).

In our work, we evaluate all four stages of 13 long-context methods, as shown in Table 1. Additionally, we list the KV cache size, pre-filling stage complexity, decoding stage complexity, and whether efficient operations are performed during the pre-filling and decoding stages for each method.

**Tri-shape Sparse Attention:** We introduce Tri-shape, a novel training-free sparse attention method that improves first-turn accuracy (Fig. 5). Unlike A-shape, which retains only the sink token and local window, Tri-shape also preserves the last window query region, forming a triangular sparse attention pattern for pre-filling. Motivated by our SCBench findings that A-shape with dense decoding improves after multiple requests, Tri-shape enhances both turn-0 and multi-request performance (§4) while maintaining LLM instruction-following ability (§G). Notably, recent concurrent work (Acharya et al., 2024) has explored similar patterns for accelerating long-context pre-filling.

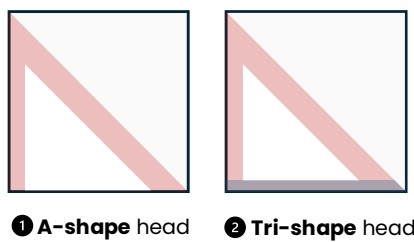

Figure 5: The sparse attention methods framework.

## 3    BENCHMARK BUILDING

SCBench features 12 tasks assessing four long-context abilities: string retrieval, semantic retrieval, global information processing, and multi-tasking, across two shared context modes—multi-turn and multi-request. These tasks span diverse domains, including code, retrieval, question answering, summarization, in-context learning, and multi-hop tracing (Fig. 2b). In total, SCBench includes 931 multi-turn sessions with 4,853 queries, averaging 5 turns per session. Task statistics are provided in Table 2, with examples and configurations in Table 3. Below, we describe the benchmark construction.

### 3.1    LONG-CONTEXT TASK DETAILS

**String Retrieval**    The core requirement for long-context LLMs is retrieving relevant information from lengthy, potentially noisy inputs. String retrieval tasks are commonly used for this evaluation (Hsieh et al., 2024; Zhang et al., 2024a). Our benchmark applies complexity analysis, inspired by algorithmic problem-solving (e.g., LeetCode), to design three tasks with varying difficulty levels.

Table 2: Overview of SCBench tasks.

| Task | Description | Capability | Avg. Input Length | Avg. Output Length | #Sessions / #Turns |
|------|-------------|------------|-------------------|--------------------|--------------------|
| Retr.KV | Key-value retrieval from many key-value pairs | String Retrieval | 125K | 943 | 100/500 |
| Retr.Prefix-Suffix | Find string with specific prefix and suffix in a dict | String Retrieval | 112K | 914 | 100/500 |
| Retr.MultiHop | Tracking variables assignment in a long input | String Retrieval | 124K | 410 | 90/450 |
| Code.RepoQA | Functions retrieval from a GitHub repo | Semantic Retrieval | 65K | 6,058 | 88/440 |
| En.QA | English Question Answering | Semantic Retrieval | 198K | 272 | 69/351 |
| Zh.QA | Chinese Question Answering | Semantic Retrieval | 1.5M | 322 | 35/189 |
| En.MultiChoice | English Multi-Choice Questions | Semantic Retrieval | 188K | 215 | 58/299 |
| Math.Find | Math computation tasks within long sequence arrays | Global Information | 120K | 172 | 100/240 |
| ICL.ManyShot | Hundreds-shot in-context learning | Global Information | 22K | 975 | 54/270 |
| En.Sum | Summarize a doc given multiple docs as input | Global Information | 104K | 1,170 | 79/350 |
| Mix.Sum+NIAH | Multi-tasking of En.Sum and Needle in A Haystack | Multi-tasking | 105K | 3,441 | 70/560 |
| Mix.RepoQA+KV | Multi-tasking of RepoQA and KV retrieval | Multi-tasking | 68K | 5,318 | 88/704 |
| **Total** | - | - | **227K** | **1,684** | **931/4,853** |

Additionally, by shifting the target string's position, we assess how well models utilize their full context window (Kamradt, 2023).

*(i) Retrieve.KV*: Given a large JSON object with numerous key-value pairs, models must accurately retrieve the value for a specified key (Liu et al., 2024d). The random KVs pose challenges for long-context LLMs, as the input is often incompressible, demanding strict $O(n)$ space for storage. This task is particularly useful for evaluating memory fuzziness in long-context methods. Each session retrieves five KV pairs, with target KVs evenly distributed across the input.

*(ii) Retrieve.Prefix-Suffix*: Given a large list of variable-length strings, models must retrieve a string matching a specified prefix and suffix. This task is particularly challenging, requiring complex functions akin to a prefix tree, with a computational cost of $O(\sum w_i^2)$, where $w_i$ is the length of the $i$-th string[1]. The presence of distractors sharing only the prefix or suffix prevents models from relying on simple lookups or induction heads (Olsson et al., 2022) for effective retrieval.

*(iii) Retrieve.MultiHop*: First introduced in RULER (Hsieh et al., 2024), this task evaluates LLMs' multi-hop tracing capabilities within long input prompts. Models must track and recall key information changes, making it ideal for testing long-context methods in KV cache reuse. Five multi-hop variable assignment chains are embedded across the context, and each test turn requires retrieving the exact multi-hop chain, i.e., all variables assigned to a specific value.

**Semantic Retrieval**   Beyond string retrieval, many real-world long-context applications require semantic understanding, such as retrieving a function from textual descriptions or answering questions from long documents. These tasks are essential in SCBench, as lossy long-context methods often struggle to abstract or comprehend information in multi-request scenarios.

*(i) Code.RepoQA*: This task requires the model to retrieve a specific function (including its name, input parameters, and full implementation) from a long source code chunk based on a precise natural language description. Unlike the original RepoQA benchmark (Liu et al., 2024c), our inputs extend to 64K tokens, with target functions evenly distributed by position within the codebase. The function descriptions were generated using GPT-4 based on the functions themselves. We also expanded the range of repositories and programming languages to include Python, C++, Java, PHP, Rust, Go, and TypeScript. Each test session involves a GitHub repository, with the model required to retrieve one function per turn across 5 turns.

*(ii) En.QA, Zh.QA, En.MultiChoice*: These tasks are extended from InfiniteBench (Zhang et al., 2024a), which provides high-quality, human-annotated QA tests based on fictional novels to eliminate external knowledge influence. The tasks require models to locate and process information from lengthy inputs, performing reasoning through aggregation or filtering. The two primary question types are: 1) Aggregation, compiling scattered information across the input (e.g., *"How much money in total did A spend on food?"*); 2) Filtering, identifying specific details from a larger set (e.g., *"What color dress did A wear when A met B for the second time?"*). In SCBench, QA pairs sharing the same input context are combined to create shared-context test sessions.

---

[1]https://leetcode.com/problems/prefix-and-suffix-search/

Table 3: Task examples and configurations in SCBench. We use different colors to highlight the questions, answers, and distractors in our examples.

| Task | Source | Configuration | Example |
|---|---|---|---|
| Retr.KV | Lost in the Middle (Liu et al., 2024d) | num kv pairs = 2500
len of key & value = 36
metric = Accuracy | Input: {<key #1>: <value #1>, ..., <key #100>: <value #100>}
Turn 1: The value of the <key #1> is? Answer 1: ...<value #1>...
Turn 2: The value of the <key #20> is? Answer 2: ...<value #20>...
Turn 3: The value of the <key #40> is? Answer 3: ...<value #40>... |
| Retr.Prefix-Suffix | Ours | size of dict = 6000
len of string = [65, 123)
metric = Accuracy | Input: Dictionary = [<str #1>, <str #2>, ..., <str #100>]
Turn 1: Prefix: <px #1>; Suffix: <sx #1>. The word with both prefix and suffix from the dict is? Answer: <str>
Turn 2: Prefix: <px #2>; Suffix: <sx #2>. Answer: <str> |
| Retr.MultiHop | RULER (Hsieh et al., 2024) | num chains = 2
num hops = 2
metric = Accuracy | Input: VAR X1 = 12345 ...... VAR Y1 = 54321 .....<noise>
VAR X2 = X1 ...... VAR Y2 = Y1 ......<noise>
VAR X3 = X2 ...... VAR Y3 = Y2 ......<noise>
Turn 1: Variables that are assigned to 12345? Answer 1: X1 X2 X3
Turn 2: Variables that are assigned to 54321? Answer 1: Y1 Y2 Y3 |
| Code.RepoQA | RepoQA (Liu et al., 2024c) | func description from GPT-4
metric = Pass@1 | Input: <func 1> + <func 2> + ... + <func 100>
Turn 1: <description of func 1>. Answer 1: <func 1>
Turn 2: <description of func 20>. Answer 2: <func 20> |
| En.QA
Zh.QA | InfiniteBench (Zhang et al., 2024a) | ground_truth from human
metric = Accuracy | Input: Read the book below and answer a question. <context>
Turn 1: <question> Be very concise. Answer 1: ...<ans>...
Turn 2: <question> Be very concise. Answer 2: ...<ans>... |
| En.MultiChoice | InfiniteBench (Zhang et al., 2024a) | ground_truth from human
metric = Accuracy | Input: Read the book and answer the question. <context>
Turn 1: <question> + <Option A,B,C,D>. Answer 1: ...<ans>...
Turn 2: <question> + <Option A,B,C,D>. Answer 2: ...<ans>... |
| Math.Find | Ours | len_array=30000
num_digits=3
metric = Accuracy | Input: <a large array of number>
Turn 1: The max number in the array is? Answer 1: ...<max number>...
Turn 2: The max number in the array is? Answer 2: ...<max number>... |
| ICL.ManyShot | ManyShotICL (Srivastava et al., 2023) | num_examples = ~150
Tasks = date, salient, tracking7
metric = Accuracy | Input: ICL Demo. 1 + Demo. 2 + ..... + Demo. 1000
Turn 1: <question>. Answer 1: ...<ans>...
Turn 2: <question>. Answer 2: ...<ans>... |
| En.Sum | Ours | Concatenated arXiv papers
ground_truth from GPT-4
num document = ~8
metric = ROUGE | Input: Doc 1 + Doc 2 + Doc 3 + ... + Doc 10.
Turn 1: Please summarize Doc 1. Answer 1: ... <summary of Doc 1>...
Turn 2: Please summarize Doc 3. Answer 2: ... <summary of Doc 3>...
Turn 3: Please summarize Doc 5. Answer 2: ... <summary of Doc 5>... |
| Mix.Sum+NIAH | Ours | num needle = 5
num document = ~8
metric = ROUGE + Acc | Input: Doc 1 + <Passkeys> + Doc 2 + ... + <Passkeys> + Doc 10.
Turn 1: Please summarize Doc 1. Answer 1: ...<summary of Doc 1>...
Turn 2: What is the needle? Answer 2: ..<needle>... |
| Mix.RepoQA+KV | Ours | num KV pairs = ~100
metric = Pass@1 + Acc | Input: <func 1> + KV pairs + <func 2> + ... + KV pairs + <func 100>
Turn 1: <description of func 1>. Answer 1: <func 1>
Turn 2: The value of the <key #1> is? Answer 2: ...<value #1>.. |

**Global Information Processing**  In addition to retrieval, some long-context tasks require leveraging and aggregating global context information, such as summarization, statistical tasks, and in-context learning (ICL) (Yu et al., 2020; Srivastava et al., 2023; Hao et al., 2022). Our benchmark includes three tasks to evaluate how well different long-context methods handle global information in multi-request.

*(i) Many-shot ICL*: We use datasets from Big-Bench Hard (Srivastava et al., 2023) to evaluate many-shot ICL capabilities. This includes three sub-tasks: *date understanding*, *salient error translation detection*, and *tracking seven shuffled objects*. Many-shot ICL contexts are shared across turns within a test session, and all sub-tasks are presented as multiple-choice questions with four options.

*(ii) Math.Find*: We extended the math find task from InfiniteBench (Zhang et al., 2024a), expanding it from finding only the maximum value to multiple statistical values. Given a large array, LLMs must find the minimum or median values, requiring effective comprehension of global context, comparisons, and statistical operations.

*(iii) En.Sum*: This task uses concatenated academic papers from *arXiv* as input, with document lengths ranging from 8K to 20K tokens. Ground truth summaries, averaging 654 tokens, were generated using GPT-4, which was prompted to produce concise one-sentence summaries for each document. The target documents for each turn are evenly distributed across the full context length.

**Multi-Tasking**  In real-world applications, LLMs often handle multiple tasks within a single session using a shared input context. For example, users may request both summarization and content retrieval simultaneously. To reflect this, SCBench includes two multi-tasking tasks:

*(i) Mix.Sum+NIAH*: This task combines document summarization with the Needle in a Haystack (Kamradt, 2023) task using a shared input prompt. A random "needle" is evenly in-

serted into the input of the En.Sum task (concatenated academic papers). The model alternates between summarization and NIAH retrieval in each test session.

*(ii) Mix.RepoQA+KV*: This task combines the RepoQA task with KV retrieval using a shared input prompt. Multiple KV pairs are evenly inserted into the RepoQA input (a long chunk of source code), including 100 KV pairs with four target KVs and the rest as distractors. The model alternates between RepoQA and KV retrieval in each test session.

## 3.2 LONG-CONTEXT SHARED CONTEXT MODES DETAILS

In addition to the carefully designed long-context tasks, we include two shared context modes—multi-turn and multi-request—to better reflect real-world applications, as shown in Fig. 2b.

*(i) Multi-turn Mode*: A typical scenario in long-context applications includes long-context chat, multi-step reasoning (e.g., Tree-of-Thought (Yao et al., 2024b)), and long-generation CoT. This mode is relevant for long-context methods with KV cache reuse, as focus shifts across turns can lead to information loss in the KV cache. Following Zheng et al. (2023a); Wang et al. (2024), we use ground-truth answers instead of model-generated content as the context for follow-up turns.

*(ii) Multi-request Mode*: Context sharing spans sessions or users, such as collaborators on a shared code repository. Models can encode shared context and reuse the KV cache across requests. Evaluating long-context methods is crucial, as some depend on the query for sparse encoding/decoding. For instance, MInference and SnapKV use the input's final segment (typically the query) to estimate the sparse pattern, assessing their generalization without query access.

## 4 EXPERIMENTS & RESULTS

**Models & Implementation Details**    We selected six open-source long-context LLMs: Llama-3.1-8B/70B (Dubey et al., 2024), Qwen2.5-72B/32B (Team, 2024), Llama-3-8B-262K (Gradient, 2024), and GLM-4-9B-1M (GLM et al., 2024), along with two gated linear models: Codestal Mamba 7B (team, 2024) and Jamba-1.5-Mini (Lieber et al., 2024). This selection covers Transformer, SSMs, and SSM-Attention Hybrid models, representing leading open-source long-context LLMs. For stability, all experiments used greedy decoding in BFloat16 on four NVIDIA A100 GPUs. We evaluated models via HuggingFace or vLLM with FlashAttention-2 (Dao, 2024) and leveraged MInference (Jiang et al., 2024) to reduce GPU memory overhead. More details on these models and infrastructure are in §D.1.

**Long-Context Method Details**    We evaluated eight long-context solution categories on our benchmark: Gated Linear RNNs (e.g., Codestral-Mamba), SSM-Attention hybrids (e.g., Jamba), sparse attention, KV cache dropping, prompt compression, KV cache quantization, retrieval, and loading, as detailed in Table 1. All methods were tested on Transformer-based long-context LLMs, except Codestral-Mamba and Jamba. We also report KV cache size, pre-filling and decoding complexity, and whether efficient operations are applied (details in §2). More details are shown in §D.2.

**Main Results**    Tables 4, 10, and Fig. 9 present the performance of various long-context methods across tasks and shared context modes in different base LLMs. Key findings include: 1) In retrieval tasks, most methods, except MInference, perform poorly, especially in exact retrieval tasks like string matching. 2) Sparse attention outperforms sparse decoding as request rounds increase, with A-shape showing the greatest improvement. Tri-shape, which adds dense bottom query tokens to A-shape, enhances first-round performance but has little effect on later rounds. It generalizes well across tasks, ranking second only to MInference. Our analysis indicates that Tri-shape improves first-turn instruction-following, boosting overall performance, while A-shape disrupts instruction information, causing random outputs (Table 17). 3) KV cache compression methods generally underperform in shared scenarios, offering only minor benefits in the first round. 4) Prompt compression improves global information tasks like many-shot ICL but significantly degrades retrieval-related performance. 5) SSM-Attention hybrids perform well in single-turn interactions but suffer accuracy drops in multi-turn tasks, especially in RepoQA and Math. Gated Linear RNN models struggle in shared context modes.

Table 4: Average performance of various long-context methods across different LLMs in two shared context modes on SCBench. For additional results on base models such as Llama-3.1-70B, Qwen2.5-32B, and Llama-3-8B-262K, see Table 10 in §E. Here, $\tau$ denotes the target compression rate.

| Methods | $\tau$ | Multi-turn Mode | | | | | Multi-request Mode | | | | |
|---|---|---|---|---|---|---|---|---|---|---|---|
| | | Retr.String | Retr.Semantic | Global | Multi-task | AVG. | Retr.String | Retr.Semantic | Global | Multi-task | AVG. |
| *LLaMA-3.1-8B* | 1 | 57.1 | 36.9 | 35.1 | 65.7 | 48.7 | 29.5 | 36.4 | 43.6 | 39.2 | 37.2 |
| ❶ A-shape | 1/32 | 14.0 | 28.9 | 31.7 | 33.7 | 27.1 | 3.2 | 33.2 | 46.3 | 27.8 | 27.6 |
| ❶ Tri-shape | 1/32 | 18.1 | 31.5 | 33.5 | 37.9 | 30.3 | 7.8 | 25.7 | 45.6 | 24.6 | 25.9 |
| ❶ MInference | 1/32 | 39.1 | 38.5 | 34.4 | 57.8 | 42.5 | 28.9 | 35.6 | 50.1 | 30.9 | 36.4 |
| ❶ LLMLingua-2 | 1/3 | 5.7 | 25.3 | 32.3 | 49.6 | 28.2 | 3.9 | 24.4 | 41.2 | 22.8 | 23.1 |
| ❷ StreamingLLM | 1/32 | 0.4 | 13.5 | 33.6 | 14.3 | 15.5 | 0.0 | 11.3 | 30.3 | 16.4 | 14.5 |
| ❷ SnapKV | 1/32 | 6.1 | 18.4 | 37.9 | 21.1 | 20.9 | 0.3 | 14.2 | 35.7 | 10.7 | 15.2 |
| ❷ PyramidKV | 1/32 | 6.3 | 18.1 | 37.1 | 22.6 | 21.0 | 0.3 | 15.1 | 34.7 | 11.0 | 15.3 |
| ❷ KIVI | 1/8 | 12.0 | 34.3 | 31.0 | 50.7 | 32.0 | 7.6 | 30.1 | 33.1 | 28.4 | 24.8 |
| ❸ CacheBlend | 1 | 56.7 | 39.4 | 35.3 | 65.6 | 49.3 | 27.6 | 35.8 | 36.2 | 39.5 | 34.8 |
| ❹ Quest | 1/32 | 8.2 | 27.3 | 33.0 | 20.1 | 22.1 | 6.7 | 25.6 | 31.8 | 14.2 | 19.6 |
| ❹ RetrievalAttention | 1/32 | 25.0 | 30.0 | 27.0 | 35.5 | 29.4 | 17.9 | 26.7 | 30.7 | 27.6 | 25.8 |
| *GLM-4-9B-1M* | 1 | 48.9 | 39.9 | 33.1 | 72.8 | 48.7 | 44.8 | 31.1 | 43.4 | 48.0 | 41.8 |
| ❶ A-shape | 1/32 | 27.2 | 31.2 | 30.7 | 58.5 | 36.9 | 20.2 | 24.1 | 40.5 | 42.6 | 31.8 |
| ❶ Tri-shape | 1/32 | 31.5 | 32.5 | 32.1 | 64.0 | 40.0 | 25.5 | 25.2 | 41.4 | 43.0 | 33.8 |
| ❶ MInference | 1/32 | 38.2 | 37.2 | 31.8 | 70.8 | 44.5 | 34.1 | 29.0 | 43.4 | 48.3 | 38.7 |
| ❶ LLMLingua-2 | 1/3 | 5.8 | 6.8 | 29.3 | 24.5 | 16.6 | 1.5 | 14.8 | 38.5 | 24.8 | 19.9 |
| ❷ StreamingLLM | 1/32 | 0.7 | 14.6 | 28.1 | 12.7 | 14.0 | 0.2 | 10.2 | 32.7 | 17.1 | 15.1 |
| ❷ SnapKV | 1/32 | 18.1 | 18.7 | 33.1 | 34.0 | 26.0 | 0.9 | 10.1 | 37.5 | 24.0 | 18.1 |
| ❷ PyramidKV | 1/32 | 18.1 | 23.9 | 30.8 | 34.9 | 26.9 | 0.6 | 4.9 | 39.8 | 21.1 | 16.6 |
| ❷ KIVI | 1/8 | 26.5 | 33.7 | 23.8 | 51.2 | 33.8 | 2.5 | 20.3 | 39.5 | 43.5 | 26.5 |
| ❹ Quest | 1/32 | 20.1 | 30.3 | 25.8 | 36.4 | 28.2 | 0.0 | 15.6 | 33.9 | 15.2 | 16.2 |
| *Qwen2.5-72B* | 1 | 51.5 | 44.4 | 38.9 | 77.0 | 52.9 | 31.1 | 46.8 | 53.0 | 52.4 | 45.8 |
| ❶ A-shape | 1/32 | 24.0 | 34.9 | 36.7 | 58.0 | 38.4 | 15.2 | 35.5 | 47.7 | 43.1 | 35.4 |
| ❶ Tri-shape | 1/32 | 25.7 | 36.6 | 37.7 | 63.8 | 40.9 | 18.6 | 38.3 | 48.5 | 44.9 | 37.6 |
| ❶ MInference | 1/32 | 45.6 | 43.5 | 38.4 | 72.8 | 50.1 | 28.6 | 44.7 | 52.2 | 52.0 | 44.4 |
| ❶ LLMLingua-2 | 1/3 | 4.2 | 28.7 | 46.2 | 27.3 | 26.6 | 2.7 | 31.2 | 49.0 | 25.8 | 27.2 |
| ❷ StreamingLLM | 1/32 | 0.7 | 16.2 | 40.2 | 18.7 | 18.9 | 0.0 | 4.2 | 4.4 | 0.0 | 2.2 |
| ❷ SnapKV | 1/32 | 3.8 | 24.3 | 43.6 | 34.1 | 26.5 | 0.0 | 6.2 | 7.0 | 0.0 | 3.3 |
| ❷ PyramidKV | 1/32 | 4.9 | 23.3 | 42.0 | 34.3 | 26.1 | 0.0 | 17.5 | 44.5 | 11.2 | 18.3 |
| ❷ KIVI | 1/8 | 12.9 | 36.7 | 49.3 | 57.4 | 39.1 | 3.0 | 40.7 | 52.8 | 46.6 | 35.8 |
| ❹ Quest | 1/32 | 5.2 | 28.3 | 41.9 | 31.5 | 26.7 | 1.2 | 24.8 | 42.1 | 15.2 | 20.8 |
| ❶ *Jamba-1.5-Mini* | - | 67.4 | 28.6 | 37.5 | 47.5 | 32.8 | 21.7 | 61.8 | 5.6 | 38.9 | 48.0 |
| ❶ *Mamba-Codestral* | - | 0.0 | 0.0 | 11 | 0.0 | 9.3 | 3.9 | 25.8 | 6.4 | 54.8 | 7.4 |

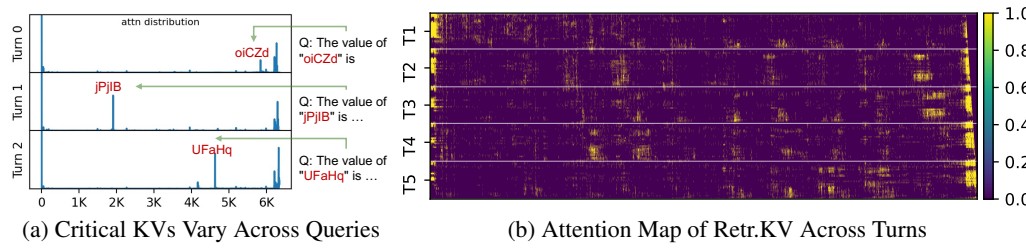

(a) Critical KVs Vary Across Queries      (b) Attention Map of Retr.KV Across Turns

Figure 6: Attention visualization of Retr.KV for the shared context across multiple turns.

## 5 ANALYSIS

**Sub-$O(n)$ Memory is *Almost* Infeasible in Multi-Turn Decoding.** Fig. 6b visualizes the attention map for the Retr.KV task across turns. While important KVs remain stable within a turn, they vary significantly between queries. This explains why $O(k)$ KV cache compression methods perform well in single-query tests but fail in follow-ups. However, the SSM-attention hybrid model Jamba shows promise in reducing memory costs by using SSM layers while maintaining $O(n)$ memory in select attention layers for future lookups (Waleffe et al., 2024). Another promising approach is CPU-GPU collaboration, where full $O(n)$ memory is stored in CPU RAM, dynamically loading relevant KVs to the GPU for sub-$O(n)$ decoding (Liu et al., 2024b). Haris & Onak (2025) theoretically proved that linear memory is required to preserve attention representational capacity, which aligns with it.

**The Sparsity in Encoding and Decoding.** We examined how sub-$O(n)$ sparse decoding struggles to maintain accuracy across multiple requests in shared context scenarios. Interestingly, sparse

methods perform well in encoding if decoding remains dense. As shown in Fig. 3a, with dense decoding ($O(n)$ memory), Tri-Shape and A-Shape achieve strong multi-request performance. While this phenomenon has been noted in single-turn tests (Sun et al., 2024c; Jiang et al., 2024), we are the first to demonstrate its potential in shared context settings. Conversely, extending sparse patterns to decoding severely degrades performance (e.g., StreamingLLM). Even with dense encoding, sparse decoding methods, particularly KV cache compression, perform poorly in shared context scenarios. This may stem from redundancy in encoding outputs, whereas decoding plays a crucial role in generation (Deng et al., 2024). Sparse encoding can still capture key information due to redundant input prompts, but sparse decoding weakens per-layer connectivity, limiting focus on critical tokens. Since sparse decoding relies on proxy tokens for global access, it constrains the formation of complex attention functions (Yun et al., 2020). We highlight the need for more advanced sparse patterns in sparse attention. Dynamic sparse attention can improve connectivity and accelerate information propagation (Jiang et al., 2024), better approximating full attention performance compared to static sparse patterns (Fig. 9).

**Compressible and Incompressible Tasks.** While $O(n)$ memory is crucial for multi-request scenarios with shared context, it can be relaxed for highly compressible inputs in simpler tasks. For example, the Needle-in-the-Haystack benchmark (Kamradt, 2023) embeds key information (the "needle") within repetitive noise (the "haystack"), allowing sub-$O(n)$ methods to achieve reasonable accuracy due to the high compressibility of noise. Similarly, summarization tasks involve compressible contexts, enabling sub-$O(n)$ methods to balance efficiency and performance. However, for dynamic and complex inputs, sub-$O(n)$ methods often fail to retain all necessary information, leading to poor performance in challenging retrieval tasks. Tasks like Retr.KV and Retr.Prefix-Suffix, which involve random and incompressible key-value pairs and strings, require models to fully utilize their context window. In summary, while compressible tasks may overestimate a model's capabilities, sub-$O(n)$ methods remain viable for simpler tasks due to their efficiency.

**Sparse Methods without Query Awareness.** A key concern with long-context methods in KV cache reuse scenarios is their reliance on the query for compression to enable efficient encoding or decoding. However, in real-world applications, a single context is often shared across multiple

Table 5: Results of query-awareness long-context methods. w/ (first) and w/o (later) query.

| LLaMA-3.1-8B | Retr.String | Retr.Semantic | Global | Multi-task |
|---|---|---|---|---|
| ❷ SnapKV | 0.0 / 0.0 | 19.0 / 9.7 | 17.9 / 14.6 | 5.1 / 0.0 |
| ❶ Tri-shape | 12.1 / 7.8 | 31.4 / 25.7 | 31.1 / 45.6 | 28.0 / 24.6 |
| ❶ MInference | 28.1 / 28.9 | 40.4 / 35.6 | 35.4 / 50.1 | 28.3 / 30.9 |

queries, requiring these methods to operate without query access. This raises the question: *Can query-dependent long-context methods generalize effectively without it?* Table 5 compares the performance of three query-aware long-context methods with and without the query, highlighting performance degradation in its absence (underlined). We found that both the KV cache compression method SnapKV and the static sparse attention method Tri-Shape struggled to maintain accuracy without the query. In contrast, the dynamic sparse attention method MInference exhibited stronger generalization, likely due to its adaptive sparse patterns, particularly its diagonal connections in the attention map.

## 6 CONCLUSION

This paper addresses a key gap in evaluating long-context methods, which have traditionally focused on single-turn interactions while overlooking shared long-context scenarios—common in real-world LLM applications. To bridge this, we introduce SCBench, a comprehensive benchmark assessing long-context methods with KV cache reuse across 12 tasks, covering string retrieval, semantic retrieval, global information processing, and multi-tasking, evaluated in two shared-context modes. Using our benchmark, we categorize long-context methods into four KV cache-centric stages: generation, compression, retrieval, and loading. We evaluate eight method categories (e.g., gated linear RNNs, hybrid models, sparse attention, KV cache dropping, quantization, retrieval, loading, and prompt compression) on eight state-of-the-art LLMs, including Llama-3.1-8B/70B, Qwen2.5-72B/32B, Llama-3-8B-262K, GLM-4-9B, Codestal Mamba, and Jamba-1.5. Our results reveal a clear KV cache management trade-off: $O(n)$ methods excel in multi-request scenarios, while sub-$O(n)$ methods perform well in single-turn but struggle with complex interactions. These findings highlight the need for evaluating long-context methods in shared-context, multi-turn scenarios, offering a more realistic benchmark and valuable insights for improving future long-context models and architectures.

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

## A RELATED WORKS

**Prefix Caching** (also known as KV cache reuse) optimizes time-to-first-token in LLM inference frameworks, particularly for shared contexts like multi-turn conversations or chatbot sessions (Zheng et al., 2024; Kwon et al., 2023; Gim et al., 2024). This technique is widely adopted by LLM providers (Gemini, 2024; Claude, 2024; OpenAI, 2024b; Azure, 2024). Recent optimizations focus on enhancing KV cache efficiency. PagedAttention (Kwon et al., 2023) reduces memory costs by partitioning the KV cache into blocks with a lookup table. HydraGen (Juravsky et al., 2024) and Cascade Inference (Ye et al., 2024) decouple attention computation for shared prefixes and unique suffixes, supporting batched multi-query kernels. RadixAttention (Zheng et al., 2024) accelerates KV lookups using a radix tree with $O(k)$ complexity and is integrated into the vLLM framework (vLLM, 2024). RAGCache (Jin et al., 2024) caches KV tensors for retrieved documents in retrieval-augmented generation, while CacheBlend (Yao et al., 2024a) improves cache utilization via partial recomputation. Despite these advancements, no existing long-context benchmarks evaluate KV cache reuse scenarios.

**Conversational and Multi-Turn Benchmarks** While multi-turn benchmarks better reflect real-world applications, many evaluations still focus on single-turn (Li et al., 2023a; Finch et al., 2023). Benchmarks like MT-Bench (Zheng et al., 2023a), ShareGPT (Domeccleston, 2023), MINT (Wang et al., 2024), MT-Bench-101 (Bai et al., 2024a), and MT-Eval (Kwan et al., 2024) assess conversational abilities, instruction-following, and complex task-solving across turns. However, they primarily focus on model consistency and information extraction rather than evaluating long-context inputs.

**Long-Context Methods of LLMs** Long-context inference faces two key bottlenecks: computational cost during pre-filling and memory cost during decoding (Fu, 2024). Pre-filling optimizations include state space models (Gu & Dao, 2024; Gu et al., 2022), linear attention methods (Peng et al., 2023; Sun et al., 2023), memory-based approaches (Munkhdalai et al., 2024), sparse attention (Jiang et al., 2024; Zhu et al., 2024; Lai et al., 2025; Yuan et al., 2025; Lu et al., 2025), hybrid techniques (Lieber et al., 2024; Ho et al., 2024; Ren et al., 2025; Fu et al., 2024; Xiao et al., 2025; Yang et al., 2025), and prompt compression (Li et al., 2023b; Jiang et al., 2023; Pan et al., 2024). Decoding optimizations focus on: 1) Attention KV reuse to reduce storage (Shazeer, 2019; Ainslie et al., 2023; Sun et al., 2024c; Liu et al., 2024a; Nawrot et al., 2024); 2) Static KV compression (Xiao et al., 2024; Han et al., 2024); 3) Dynamic KV compression, including cache discarding (Zhang et al., 2024b; Ge et al., 2024; Liu et al., 2023; Li et al., 2024c) and offloading (Ribar et al., 2024; Tang et al., 2024; Dai et al., 2024; Liu et al., 2024b; Chen et al., 2025; Sun et al., 2024a; Hooper et al., 2024; Desai et al., 2024); 4) Hierarchical speculative decoding (Sun et al., 2024b). Most methods are tested on single-turn benchmarks and employ query-conditioned lossy techniques, which may

degrade performance in multi-turn scenarios with prefix caching. This limitation motivates the design of SCBench, a benchmark that evaluates long-context solutions in shared context settings.

## B  COMPARED TO PRIOR LONG-CONTEXT BENCHMARK

We have compared SCBench against existing long-context benchmarks across long-context capability assessed, request types considered, and implementation they adopted, as shown in Table 6.

Table 6: Comparison of Long-Context Benchmarks.

| | Long-Context Capability | | | | Request Type | | | Implementation |
|---|---|---|---|---|---|---|---|---|
| | Precise Retrieval | Semantic Retrieval | Global Information | Multi-Tasking | Single Question | Multi-Turn | Multi-Request | KV Cache Reuse |
| LongBench (Bai et al., 2024b) | | ✓ | ✓ | | ✓ | | | |
| InfiniteBench (Zhang et al., 2024a) | ✓ | ✓ | ✓ | | ✓ | | | |
| RULER (Hsieh et al., 2024) | ✓ | ✓ | ✓ | | ✓ | | | |
| LongCTXBench (Yuan et al., 2024) | ✓ | ✓ | ✓ | | ✓ | | | |
| HELMET (Yen et al., 2024) | ✓ | ✓ | ✓ | | ✓ | | | |
| Michelangelo (Vodrahalli et al., 2024) | ✓ | ✓ | | | ✓ | | | |
| SCBench | ✓ | ✓ | ✓ | ✓ | ✓ | ✓ | ✓ | ✓ |

We also directly compare the testing results of long-context methods on prior benchmarks and SCBench to show the unique insights our benchmark provides. We mainly compare two common long-context capability: summarization (as shown in Table 7), and retrieval (as shown in 8). The summarization sub-tasks we used is En.Sum for InfiniteBench (Zhang et al., 2024a), and gov-report for LongBench (Bai et al., 2024b). The retrieval sub-tasks we used is Retr.KV for InfiniteBench (Zhang et al., 2024a), and Passage-retrieval for LongBench (Bai et al., 2024b). In addition, LongCTXBench (Yuan et al., 2024) also analyzes the performance boundaries of long-context efficient methods from a KV-cache-centric perspective. However, it does not consider multi-request scenarios and only focuses on the KV Cache Generation and Compression stages.

Table 7: Comparing the summarization capability of efficient long-context methods on prior benchmarks and our SCBench.

| | Prior Benchmarks | | SCBench | | | | | |
|---|---|---|---|---|---|---|---|---|
| Model | InfiniteBench | LongBench | Multi Request | Turn-1 | Turn-2 | Turn-3 | Turn-4 | Turn-5 |
| *Llama-3.1-8B-Inst* | 28.5 | 36.6 | 38.3 | 44.2 | 42.1 | 35.8 | 37.6 | 42.3 |
| A-Shape | 24.5 | 33.5 | 28.8 | 26.1 | 30.8 | 33.8 | 40.8 | 40.4 |
| Tri-Shape | 27.4 | 33.9 | 30.2 | 32.1 | 30.0 | 34.0 | 41.0 | 40.3 |
| Minference | 28.9 | 33.9 | 36.7 | 40.6 | 36.1 | 39.7 | 43.5 | 43.7 |
| StreamingLLM | 27.3 | 32.0 | 30.2 | 29.4 | 26.1 | 27.7 | 27.3 | 26.9 |
| SnapKV | 28.3 | 33.2 | 29.9 | 36.2 | 29.4 | 28.6 | 28.1 | 31.0 |
| LLMLingua | 23.1 | 32.0 | 30.1 | 32.5 | 22.5 | 26.6 | 25.7 | 26.6 |

Table 8: Comparing the retrieval capability of efficient long-context methods on prior benchmarks and our SCBench.

| | Prior Benchmarks | | SCBench | | | | | |
|---|---|---|---|---|---|---|---|---|
| Model | InfiniteBench | LongBench | Multi Request | Turn-1 | Turn-2 | Turn-3 | Turn-4 | Turn-5 |
| *Llama-3.1-8B-Inst* | 57 | 100 | 56 | 62 | 59 | 68 | 66 | 70 |
| A-Shape | 0 | 42 | 3 | 0 | 12 | 22 | 28 | 33 |
| Tri-Shape | 21 | 100 | 5 | 14 | 19 | 25 | 32 | 38 |
| Minference | 33 | 100 | 14 | 31 | 35 | 46 | 56 | 50 |
| StreamingLLM | 0 | 84 | 0 | 2 | 1 | 0 | 0 | 0 |
| SnapKV | 4 | 100 | 0 | 0 | 0 | 0 | 0 | 0 |
| LLMLingua | 0 | 90 | 0 | 0 | 1 | 2 | 0 | 0 |

We found SCBench can better identify the weakness of long-context methods under the KV cache reuse scenarios, such as the general incapability of KV cache compression methods on multi-request

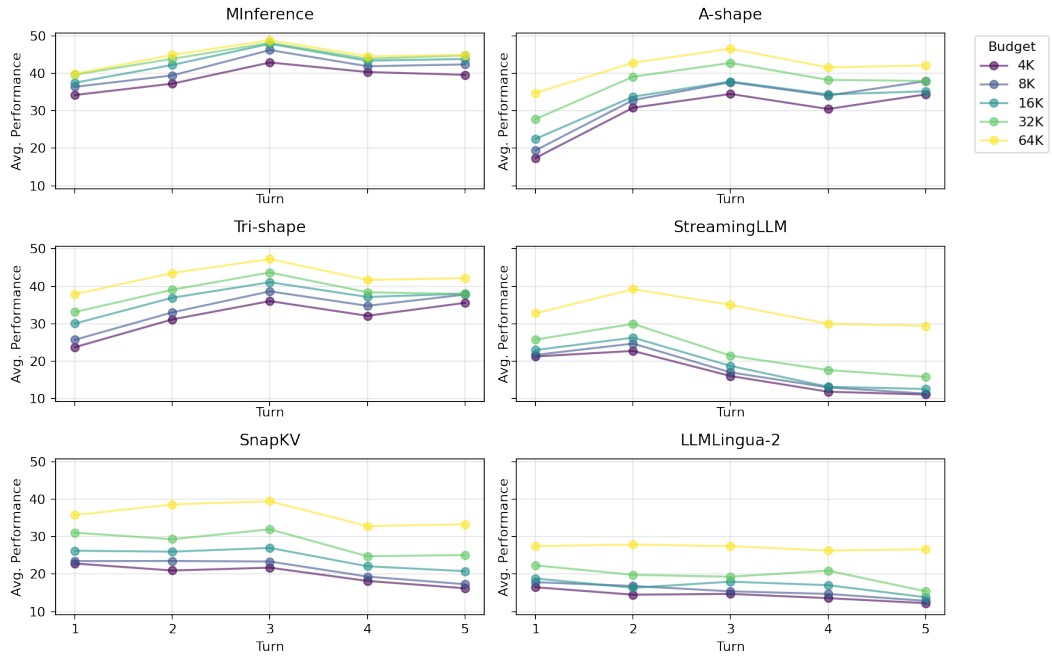

Figure 7: Hyper-parameters analysis: averaged performance of efficient long-context methods with different computing budgets under the multi-turn mode of SCBench. The input length is 128K, meaning that 4K, 8K, 16K, 32K, and 64K correspond to sparsity budgets of 1/32, 1/16, 1/8, 1/4, and 1/2, respectively.

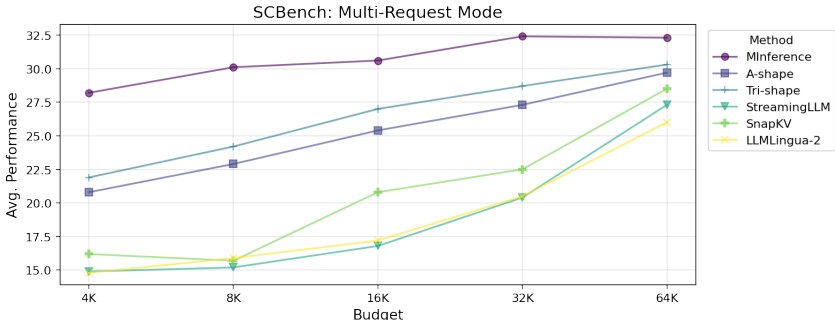

Figure 8: Hyper-parameters analysis: averaged performance of efficient long-context methods with different computing budgets under the multi-request mode of SCBench. The input length is 128K, meaning that 4K, 8K, 16K, 32K, and 64K correspond to sparsity budgets of 1/32, 1/16, 1/8, 1/4, and 1/2, respectively.

mode and follow-up queries in the multi-turn mode, as well as the increasing accuracy of sparse attention under multi-turn mode.

## C  HYPER-PARAMETERS OF EFFICIENT LONG-CONTEXT METHODS

We conduct extensive experiments with various computing budgets for the efficient long-context methods we covered. The results are shown in Figure 7 and Figure 8 for the multi-turn mode and multi-request mode respectively.

From the results, we can derive the following insights: 1) Most methods show minimal performance degradation at a 1/2 budget (e.g., A-shape and Tri-shape drop by 5-6 points, SnapKV drops by 11 points). However, as sparsity increases, performance declines significantly. For example,

StreamingLLM and SnapKV drop by 26 and 19 points, respectively, under a 1/4 budget. 2) More accurate sparse methods can maintain performance even under higher sparsity. For instance, MInference achieves performance at a 1/32 budget comparable to A-shape and Tri-shape at a 1/4 budget. 3) While some methods exhibit similar performance in single-turn scenarios, they diverge significantly in multi-turn and multi-request scenarios. For example, SnapKV outperforms StreamingLLM in turn-1 but performs significantly worse in turn-2. In some tasks, changing the budget has little impact on turn-1 performance but substantially affects turn-2 and subsequent turns, such as in Long Document QA tasks and summarization.

## D    EXPERIMENT DETAILS

### D.1    LONG-CONTEXT METHODS DETAILS

This section will introduce the long-context methods (as shown in Table 1) that involved in our paper.

**State Space Models (SSMs)**    are powerful models often used for modeling dynamic systems, particularly in time series analysis, control theory, and machine learning. As language are naturally time series data, recent advancements have integrated SSMs into language modeling architectures, showcasing their potential as alternatives to traditional models like RNNs and Transformers. Due to their linear complexity, they are especially suitable for long sequence tasks. For instance, models such as S4 (Hasani et al., 2023) and Mamba (Gu & Dao, 2024) have demonstrated superior efficiency in handling sequential data with reduced computational complexity compared to their predecessors and comparable accuracy in tasks such as language modeling. However, SSMs were also criticized for their reduced memorization capability and their limited capability in copy-pasting (Jelassi et al., 2024).

**Mamba-Attention Hybrid Architecture**    interleaves blocks of Transformers and Mamba layers, aiming to obtain the benefits of both architecture, i.e., the expressive power of Transformer and the linear complexity of Mamba layers. Jamba (Lieber et al., 2024) and Samba (Ren et al., 2025) are representative efforts on this direction. Waleffe et al. (2024) also highlights the potential of such hybrid architectures and found only a few number of attention layers can lead to significant performance increase compared to pure SSMs models.

**Sparse Attention**    is extensively studied for long sequence processing, including image synthesis and multi documents question answering. We test three sparse attention approach in our paper: A-shape, Tri-shape, and MInference. In A-shape attention, each token is only allowed in to attend to initial tokens and local tokens, resulting a A-shape on its attention map (Xiao et al., 2024). Tri-shape attention is a variant of A-shape method, we introduced in our paper, where we add a dense attention space at the bottom of the triangle A-shape attention matrix. This is based on the promising results of sparse encoding with dense decoding, where the dense space we added is a natural extrapolate of the dense decoding idea. MInference (Jiang et al., 2024) is the state-of-the-art dynamic sparse attention approach where the exact sparse pattern are dynamically built on-the-fly to better approximate full attention operation.

**KV Cache Compression**    is a series of studies that attempt to solve the linearly growing memory (often referred as KV Cache) cost in LLMs inference. For example, StreamingLLM (Xiao et al., 2024) use a constant size of KV Cache in their decoding steps, where only the state of initial and local tokens are preserved, and the rest part of KV Caches are evicted from the memory. SnapKV (Li et al., 2024c) introduces the concept of the observation window. It selects the top-K KVs that are extensively attended to in the observation window, and removes other KVs from the Cache. This method was reported to performance well in simple Neeld-in-A-Haysatck tasks and many other natural language tasks.

**KV Cache Quantization**    aims to reduce the memory footprint of KV cache via quantization. KIVI (Liu et al., 2024e) employed a per-channel quantization for the Key tensor, and a per-token quantization for Value tensor. In our evaluation, we use a 2 bit algorithms with group size of 32 an residual length of 32.

**KV Cache Retrieval**   indicates the operation to retrieve pre-cached KV cache and reuse them for incoming requests. Most of the frameworks employ an exact match algorithm, i.e., only retrieve and reuse the KV cache is the the shared context match exactly. However, there also exists approach such as CacheBlend (Yao et al., 2024a) that retrieve KV cache once the input is similar enough semantically with one former request from the cache base.

**KV Cache Loading**   Due to the huge memory result from the long-context KV cache, researchers have proposed novel approaches that make use of the extensive CPU RAM and load only partial of the KV cache to GPU per-token for more efficient decoding. For example, Quest (Tang et al., 2024) estimates the importance of Keys in a page granularity, and only the topK important Keys and corresponding Values are loaded to CUDA HBM for the attention computation. Retrieval Attention (Liu et al., 2024b) constructs vector database on CPU RAM to find the topK critical Keys efficiently. In additional, it tailored a pipeline for decoding to hide the memory movement from CPUs to GPUs.

**Prompt Compression**   aims to compress the prompt to obtain a more compact representation of the input before send it to the LLMs (Li et al., 2023b; Jiang et al., 2023). LLMLingua-2 (Pan et al., 2024) is a supervised model that assess the importance of individual token as a token classification task. It was shown in provide up to 20x compression on many tasks, with only minimal performance sacrifice.

## D.2   ADDITIONAL IMPLEMENTATION DETAILS

Table 9: Configurations of long-context methods in SCBench.

|  | Methods | Configurations |
|---|---|---|
| SSMs | Mamba-Codestral-7B-v0.1 | chunk size: 256, conv kernel: 4, expand: 2, head dim: 64, hidden size: 4096, intermediate size: 8192, n groups: 8, norm before gate: true, num heads: 128, num hidden layers: 64, state size: 128 |
| Hybrid Models | AI21-Jamba-1.5-Large | num hidden layers: 72, hidden size: 8192, intermediate size: 24576, num attention heads: 64, num key value heads: 8, mamba d state: 16, mamba d conv: 4, mamba expand: 2, mamba conv bias: true, num experts: 16, num experts per tok: 2, attention:mamba = 1:7, number layers per block: 8 |
| Sparse Attention | Tri-Shape | num local: 4096, num initial: 128, num dense rows: 128 |
|  | A-Shape | num local: 4096, num initial: 128 |
|  | MInference | Pattern search data: KV retrieval a-shape: 1024/4096 vertical-slash: 30/2048, 100/1800, 500/1500, 3000/200 block-sparse: 100 blocks |
| KV Cache Compression | StreamingLLM | num local: 4096, num initial: 128 |
|  | PyramidKV | window size: 32, max capacity prompt: 4096, kernel size: 5, pooling: avgpool |
|  | SnapKV | window size: 32, max capacity prompt: 4096, kernel size: 5, pooling: avgpool |
| KV Cache Quantization | KIVI | bit size: 2; group size: 32 residual length: 32 |
| KV Cache Retrieval | CacheBlend | recompute ratio: 15%, chunk size: 512 |
| KV Cache Loading | Quest | chunk size: 16; token budget: 4096 |
|  | RetrievalAttention | topk: 2000; index type: IVF index |
| Prompt Compression | LLMLingua-2 | compression rate: 0.333 |

In Table 9, we report the configuration we used for the long-context methods we involved in our experiments. For the Mamba-Codestral-7B-v0.1 model and AI21-Jamba-1.5-Large, we report the architecture details of other models. For SSMs models, the state size and number of layers are crucial properties, as all previous information are compressed and saved in this fixed size of states. Moreover, the number of groups and number heads are also important as they implement channel mixing which

Table 10: The average results of various long-context methods on Llama-3.1-70B, Qwen2.5-32B, and Llama-3-8B-262K with two shared context modes on SCBench.

| Methods | Multi-turn Mode | | | | | Multi-request Mode | | | | |
|---|---|---|---|---|---|---|---|---|---|---|
| | Retr.String | Retr.Semantic | Global | Multi-task | AVG. | Retr.String | Retr.Semantic | Global | Multi-task | AVG. |
| *Llama-3-8B-262K* | 29.2 | 33.3 | 26.7 | 63.5 | 38.2 | 17.1 | 30.0 | 25.5 | 34.1 | 26.7 |
| A-shape | 9.9 | 27.2 | 25.6 | 55.6 | 29.6 | 7.8 | 27.3 | 22.0 | 35.2 | 23.1 |
| Tri-shape | 11.1 | 29.6 | 26.3 | 60.6 | 31.9 | 8.2 | 22.4 | 22.5 | 35.9 | 22.3 |
| MInference | **17.5** | **33.5** | **26.7** | **66.0** | **36.2** | **8.3** | **32.1** | 25.6 | **40.0** | **26.5** |
| StreamingLLM | 0.5 | 12.6 | 22.6 | 10.1 | 11.4 | 0 | 1.0 | 22.6 | 0.1 | 5.9 |
| SnapKV | 0.5 | 4.2 | 21.9 | 0.5 | 6.7 | 0.0 | 1.1 | 24.5 | 0.1 | 6.4 |
| LLMLingua-2 | 3.4 | 21.0 | 24.5 | 23.0 | 18.0 | 3.9 | 24.4 | **42.4** | 22.6 | 23.3 |
| *Llama-3.1-70B* | 20.9 | 45.4 | 45.7 | 70.3 | 45.6 | 3.1 | 47.9 | 48.1 | 47.8 | 36.7 |
| A-shape | 4.8 | 34.7 | 40.5 | 26.9 | 26.7 | 3.2 | 35.7 | 46.3 | 33.8 | 29.7 |
| Tri-shape | 6.7 | 37.1 | 42.0 | 31.1 | 29.2 | 3.8 | 40.5 | 46.5 | 34.2 | 31.2 |
| MInference | **19.5** | **42.5** | **43.1** | **65.6** | **42.4** | **7.3** | **43.7** | **48.2** | **46.1** | **36.3** |
| StreamingLLM | 0.2 | 6.4 | 22.8 | 3.7 | 8.3 | 0.0 | 10.9 | 31.2 | 0.0 | 10.5 |
| SnapKV | 0.7 | 3.7 | 25.0 | 1.5 | 7.7 | 0.1 | 14.0 | 36.9 | 0.0 | 12.8 |
| LLMLingua-2 | 6.7 | 38.8 | 38.7 | 31.0 | 28.8 | 4.5 | 32.0 | 38.6 | 26.7 | 25.5 |
| *Qwen2.5-32B* | 46.8 | 42.6 | 40.6 | 73.4 | 50.9 | 25.0 | 44.5 | 55.3 | 49.9 | 43.7 |
| A-shape | 15.0 | 33.8 | 38.7 | 59.5 | 36.7 | 9.6 | 34.1 | 53.7 | 38.6 | 34.0 |
| Tri-shape | 18.5 | 34.6 | 40.4 | 64.0 | 39.4 | 11.7 | 37.4 | **56.4** | 41.1 | 36.7 |
| MInference | **35.4** | **39.9** | **40.8** | 69.9 | **46.5** | **17.7** | **42.7** | **56.4** | **48.6** | **41.4** |
| StreamingLLM | 0.2 | 4.3 | 8.4 | 6.3 | 4.8 | 0.0 | 1.8 | 7.4 | 0.0 | 2.3 |
| SnapKV | 3.3 | 3.9 | 27.1 | 1.5 | 9.0 | 0.0 | 4.9 | 9.8 | 0.0 | 3.7 |
| LLMLingua-2 | 3.4 | 28.2 | 38.9 | 26.9 | 24.3 | 2.7 | 26.6 | 36.5 | 22.4 | 22.1 |

shown to be critical for the expressive power. For Mamba-Attention hybrid architecture, the present the ratio of attention layers and mamba layers. As the Jamba model is also a MoE, we also represent the number of experts and the number of experts activated per token.

In Sparse Attention, we report the the local size and initial size of tokens that Tri-shape and A-shape can attend to. For Tri-shape, we add a dense space of size 64 at the bottom of the attention matrix. MInference is a dynamic sparse attention, where the exact sparse patterns are built conditioned on the inputs. According to Jiang et al. (2024), we search the sparse patterns for attention heads with the task of KV retrieval, and we also report the search space (i.e., the distribution of sparse index) for the exact pattern. In KV Cache compression, we report the composition of KV used in StreamingLLM. The observation window and max capacity of KV Cache size, the kernel size used to identify top-k KVs are reported in the Table. For KV cache quantization, retrieval and loading, we use the default hyper-parameters in their original implementation and reported them in Table 9.

We use tensor parallel when testing models larger than 7B parameters, with 8*A100 40GB machines or 4*H100 80GB machines. Specifically, we use our customized A-shape, Tri-shape, and MInference kernels in sparse attention testing, utilizing PIT (Zheng et al., 2023b) with FlashAttention (Dao, 2024) implemented on Triton (Tillet et al., 2019). `vLLM-0.5`[2] is used as the inference framework in our testing, and the `flash_attn-2.5` kernels were overwritten with our own kernels. For KV Cache compression, our implementation is based on the huggingface implementation of SinkCache for StreamingLLM[3], and official implementation of [4]. For SSMs and Mamba-Attention Hybrid models, we use the triton version of mamba[5] kernels together with `causal-conv1d-1.4`[6]. For prompt compression, we use the official implementation of LLMLinugua-2[7] to compressed the prompt first then use vLLM for further inference.

# E   ADDITIONAL EXPERIMENT RESULTS

The results for Llama-3.1-70B, Qwen2.5-32B, and Llama-3-8B-262K are shown in Table 10.

---

[2] `https://github.com/vllm-project/vllm`
[3] `https://huggingface.co/docs/transformers/main/en/kv_cache#sink-cache`
[4] `https://github.com/FasterDecoding/SnapKV`
[5] `https://github.com/state-spaces/mamba`
[6] `https://github.com/Dao-AILab/causal-conv1d`
[7] `https://github.com/microsoft/LLMLingua`

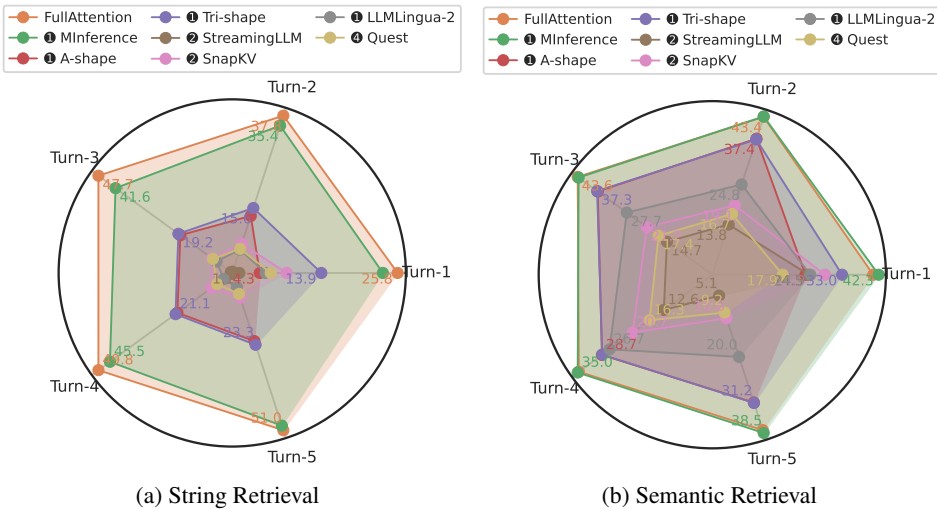

(a) String Retrieval   (b) Semantic Retrieval

Figure 9: Performance of different long-context methods across various tasks and turns. The results for multi-tasking tasks are shown in Fig. 10, and the results are averaged across all tested base LLMs.

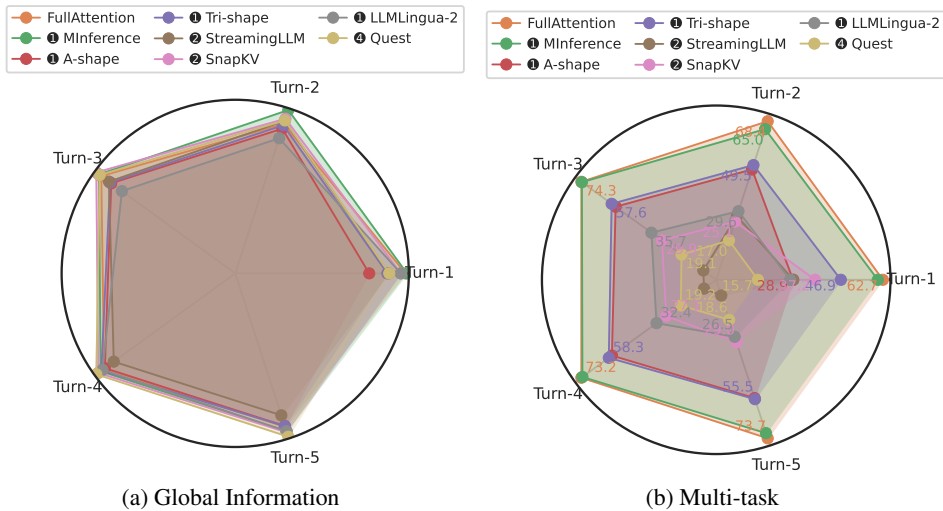

(a) Global Information   (b) Multi-task

Figure 10: Performance of different long-context methods across various turns in Global Information and Multi-tasking tasks on SCBench. The results are averaged across all tested base LLMs.

We can found similar the following key insights from Table 10. MInference consistently outperforms other approaches across tasks, particularly in multi-turn mode, demonstrating strong results in both retrieval and multi-task scenarios. Sparse attention methods like A-shape and Tri-shape show promise, with Tri-shape excelling in multi-request mode due to its integration of bottom query tokens, which boosts first-turn performance and improves instruction-following. However, Tri-shape's advantage decreases slightly in multi-task settings, although it still ranks second overall. KV cache compression methods underperform in shared contexts, offering minimal gains, especially in retrieval and global information tasks, with SnapKV showing particularly poor results. Prompt compression methods perform well in tasks requiring global context, such as many-shot ICL, but struggle significantly in retrieval tasks, leading to performance degradation. Meanwhile, StreamingLLM and SnapKV consistently deliver the weakest results, particularly in multi-turn mode, indicating they are not well-suited for long-context tasks with repeated requests. Overall, methods like Tri-shape and MInference, which combine sparse attention and efficient token management, demonstrate the most consistent improvements, while compression-focused approaches show limited effectiveness in more dynamic or retrieval-heavy tasks.

Table 11: The results breakdown of SCBench for all sub-tasks in multi-turn mode.

| Methods | Retr.KV | Retr.PS | Math.Find | RepoQA | En.QA | Zh.QA | En.MC | ICL | EN.Sum | Math | Mix.Sum +NIAH | Mix.RepoQA +KV |
|---|---|---|---|---|---|---|---|---|---|---|---|---|
| *GLM-4-1M* | 49.0 | 39.2 | 58.6 | 60.5 | 33.6 | 15.2 | 50.2 | 47.0 | 37.8 | 14.4 | 67.4 | 78.2 |
| MInference | 51.2 | 28.6 | 34.8 | 53.4 | 33.3 | 15.0 | 49.3 | 47.0 | 37.7 | 10.6 | 68.3 | 73.4 |
| A-shape | 25.2 | 42.4 | 14.0 | 42.3 | 28.1 | 14.1 | 42.4 | 49.6 | 32.9 | 9.6 | 65.3 | 51.8 |
| Tri-shape | 32.2 | 47.8 | 14.4 | 44.8 | 28.1 | 15.6 | 44.1 | 50.0 | 34.1 | 12.2 | 64.6 | 63.4 |
| StreamingLLM | 0.0 | 0.0 | 0.0 | 0.2 | 5.2 | 2.0 | 32.2 | 70.0 | 5.8 | 3.0 | 12.7 | 0.0 |
| SnapKV | 0.2 | 0.0 | 26.0 | 0.5 | 13.9 | 2.4 | 34.2 | 70.0 | 6.5 | 7.2 | 41.6 | 0.9 |
| LLMLingua-2 | 0.0 | 1.6 | 15.8 | 2.0 | 5.2 | 3.5 | 20.1 | 45.6 | 32.8 | 9.4 | 48.2 | 0.9 |
| *Llama-3.1-8B* | 80.8 | 42.8 | 47.6 | 40.4 | 29.3 | 21.1 | 57.0 | 42.6 | 40.7 | 22.0 | 60.7 | 70.7 |
| MInference | 70.8 | 15.6 | 30.8 | 48.2 | 30.1 | 22.5 | 57.9 | 49.3 | 39.8 | 14.2 | 56.3 | 59.3 |
| A-shape | 17.8 | 5.6 | 18.5 | 34.3 | 21.2 | 17.1 | 46.2 | 48.1 | 33.4 | 13.4 | 50.8 | 16.6 |
| Tri-shape | 24.2 | 7.0 | 23.2 | 34.5 | 24.6 | 20.6 | 50.5 | 48.1 | 35.2 | 17.2 | 50.8 | 25.0 |
| StreamingLLM | 0.2 | 0.0 | 0.1 | 0.5 | 8.7 | 10.5 | 39.1 | 68.9 | 27.2 | 9.6 | 28.9 | 0.5 |
| SnapKV | 0.0 | 0.0 | 0.0 | 0.0 | 1.7 | 1.9 | 17.7 | 42.6 | 3.4 | 4.2 | 4.1 | 0.0 |
| LLMLingua-2 | 0.0 | 1.6 | 15.4 | 2.0 | 23.5 | 23.0 | 61.5 | 50.4 | 35.4 | 11.2 | 49.6 | 49.6 |
| *Llama-3-8B* | 24.0 | 15.8 | 47.8 | 41.8 | 29.0 | 13.8 | 53.1 | 30.7 | 37.5 | 11.8 | 67.0 | 60.0 |
| MInference | 16.0 | 3.6 | 32.8 | 42.5 | 30.1 | 12.3 | 53.5 | 32.6 | 37.0 | 10.4 | 68.6 | 63.4 |
| A-shape | 2.2 | 2.0 | 25.4 | 32.3 | 21.7 | 12.7 | 46.6 | 32.2 | 32.8 | 11.8 | 64.8 | 46.4 |
| Tri-shape | 3.4 | 3.8 | 26.2 | 33.4 | 24.1 | 12.8 | 48.3 | 33.3 | 32.7 | 12.8 | 65.3 | 55.9 |
| StreamingLLM | 0.8 | 0.0 | 0.7 | 0.0 | 9.6 | 1.3 | 48.8 | 58.9 | 4.2 | 4.6 | 20.1 | 0.0 |
| SnapKV | 0.0 | 0.0 | 1.4 | 0.0 | 2.1 | 1.4 | 17.7 | 62.2 | 2.1 | 1.4 | 0.9 | 0.0 |
| LLMLingua-2 | 0.0 | 0.4 | 9.8 | 1.1 | 21.2 | 13.4 | 57.2 | 34.8 | 31.6 | 7.2 | 45.9 | 0.2 |
| *Llama-3.1-70B* | 27.2 | 1.6 | 33.8 | 67.0 | 35.4 | 20.7 | 62.2 | 58.5 | 41.2 | 37.4 | 62.1 | 78.4 |
| MInference | 28.0 | 1.0 | 29.4 | 60.2 | 33.0 | 23.3 | 57.4 | 54.4 | 39.8 | 35.2 | 52.1 | 77.0 |
| A-shape | 1.2 | 0.0 | 13.2 | 50.0 | 27.0 | 18.0 | 46.7 | 52.2 | 36.5 | 32.8 | 35.3 | 18.6 |
| Tri-shape | 2.8 | 0.2 | 17.1 | 50.5 | 28.0 | 18.7 | 55.5 | 55.6 | 37.0 | 33.4 | 38.3 | 23.9 |
| StreamingLLM | 0.0 | 0.0 | 0.5 | 0.4 | 6.0 | 0.8 | 23.0 | 61.1 | 3.6 | 3.8 | 7.0 | 0.3 |
| SnapKV | 0.0 | 0.0 | 2.2 | 0.0 | 1.3 | 1.5 | 14.9 | 64.5 | 1.9 | 8.8 | 2.2 | 0.7 |
| LLMLingua-2 | 0.0 | 4.2 | 16.0 | 31.6 | 33.6 | 22.5 | 74.7 | 56.7 | 37.1 | 22.2 | 1.4 | 60.6 |
| *Qwen2.5-72B* | 40.8 | 62.2 | 51.5 | 65.5 | 40.0 | 10.9 | 65.7 | 66.7 | 37.9 | 12.2 | 71.9 | 82.0 |
| MInference | 43.4 | 46.4 | 47.0 | 59.3 | 41.2 | 11.4 | 67.0 | 64.1 | 38.2 | 12.8 | 72.0 | 73.6 |
| A-shape | 17.4 | 32.0 | 22.7 | 45.9 | 31.9 | 12.8 | 52.7 | 64.4 | 33.7 | 12.0 | 69.4 | 46.6 |
| Tri-shape | 21.0 | 31.4 | 24.8 | 48.0 | 32.8 | 12.6 | 57.5 | 64.8 | 35.8 | 12.4 | 70.0 | 57.5 |
| StreamingLLM | 0.0 | 0.0 | 1.2 | 0.2 | 3.8 | 0.9 | 63.9 | 19.3 | 3.9 | 0.0 | 14.5 | 0.5 |
| SnapKV | 0.0 | 0.0 | 3.4 | 0.0 | 0.3 | 1.0 | 70.8 | 34.2 | 2.0 | 0.0 | 2.7 | 0.5 |
| LLMLingua-2 | 0.0 | 3.2 | 9.3 | 4.3 | 32.5 | 14.7 | 73.8 | 72.2 | 33.1 | 33.2 | 53.8 | 0.9 |
| *Qwen2.5-32B* | 56.4 | 39.4 | 44.7 | 64.5 | 37.1 | 6.0 | 68.3 | 75.9 | 35.5 | 10.4 | 69.8 | 77.0 |
| MInference | 27.8 | 27.8 | 50.6 | 57.5 | 34.5 | 8.0 | 65.3 | 76.3 | 35.8 | 10.4 | 70.7 | 69.1 |
| A-shape | 14.4 | 13.6 | 16.9 | 46.4 | 30.1 | 4.6 | 54.1 | 76.7 | 30.6 | 8.8 | 67.2 | 51.8 |
| Tri-shape | 18.2 | 16.6 | 20.8 | 47.3 | 30.1 | 6.8 | 59.6 | 76.3 | 33.8 | 11.2 | 68.2 | 59.8 |
| StreamingLLM | 0.0 | 0.0 | 0.7 | 0.4 | 3.3 | 0.0 | 17.0 | 21.1 | 3.6 | 0.6 | 12.5 | 0.1 |
| SnapKV | 0.0 | 0.0 | 10.0 | 0.0 | 0.3 | 0.8 | 18.1 | 37.4 | 2.3 | 41.6 | 2.7 | 0.4 |
| LLMLingua-2 | 0.0 | 4.0 | 6.2 | 6.7 | 31.4 | 15.9 | 66.7 | 66.7 | 29.5 | 20.4 | 52.7 | 1.1 |
| *Jamba-1.5-Mini* | 67.4 | 28.6 | 37.5 | 47.5 | 32.8 | 21.7 | 61.8 | 38.9 | 48.0 | 5.6 | 71.0 | 71.6 |
| *Codestral-Mamba* | 0.0 | 0.0 | 0.4 | 0.0 | 5.7 | 5.1 | 21.8 | 33.3 | 18.0 | 4.0 | 12.4 | 0.0 |

In Table 11 showcases the performance of various methods across a range of tasks, including retrieval (Retr.KV, Retr.PS), QA (En.QA, Zh.QA), summarization (En.Sum), code understanding and function retrieval (RepoQA), math, and in-context learning (ICL). Each method demonstrates varying strengths and weaknesses across these domains.

**Retrieval tasks** (Retr.KV, Retr.PS), which test exact information retrieval ability, are dominated by methods such as GLM-4-1M and MInference. GLM-4-1M consistently performs well in these tasks, with Retr.KV at 49.0 and Retr.PS at 39.2. MInference also demonstrates strong performance in retrieval, particularly with a score of 51.2 in Retr.KV. However, methods like StreamingLLM and SnapKV show almost no retrieval capability, with near-zero scores, indicating poor handling of exact information recall.

For **natural language tasks** like QA (En.QA, Zh.QA) and summarization (EN.Sum), we see a different pattern. GLM-4-1M and Qwen2 models excel in these areas, particularly in English and Chinese QA tasks. For example, Qwen2-72B achieves scores of 40.0 in En.QA and 66.7 in EN.Sum, indicating strong natural language processing abilities. MInference also performs well but is slightly behind GLM-4-1M and Qwen2, with comparable scores. Interestingly, methods like Tri-shape and A-shape show moderate performance in QA but underperform in summarization tasks compared to the top performers.

In **code understanding tasks** (RepoQA), GLM-4-1M leads with a score of 60.5, followed by Qwen2-72B at 65.5, demonstrating strong capabilities in handling structured language and retrieving functional information. Methods like MInference (53.4) and Tri-shape (44.8) perform moderately well, while StreamingLLM and SnapKV are almost ineffective, scoring near zero. This suggests that StreamingLLM and SnapKV struggle with code-related tasks requiring structured reasoning.

In **math tasks**, MInference and GLM-4-1M are the top performers, with scores of 34.8 and 58.6, respectively, showing proficiency in handling mathematical reasoning. However, methods like Tri-shape and A-shape struggle in math tasks, indicating that these sparse attention mechanisms may not generalize well to numerical reasoning. StreamingLLM and SnapKV again show little to no ability in math, with minimal scores across the board.

Finally, in **in-context learning** tasks, where the model's ability to generalize and adapt is tested, GLM-4-1M and Qwen2 models stand out. Qwen2-72B achieves a high score of 66.7, while GLM-4-1M also scores well at 47.0, indicating strong adaptability. MInference, Tri-shape, and A-shape show moderate ICL performance, but methods like SnapKV and LLMLingua-2 lag significantly, reflecting their limited generalization capabilities in ICL.

Overall, GLM-4-1M and MInference consistently perform well across most tasks, especially in retrieval, QA, and ICL, with the Qwen2 models also excelling in natural language processing and in-context learning. Sparse attention methods like A-shape and Tri-shape show moderate performance in specific areas, while methods like StreamingLLM and SnapKV consistently underperform across the board, particularly in tasks requiring retrieval and code understanding.

In Table 12, we present the results breakdown for the multi-request mode. Comparing the performance across multi-turn and multi-request modes, we found the following key differences, particularly in retrieval tasks. In multi-turn mode, methods like GLM-4-1M and MInference demonstrate strong retrieval capabilities, with high scores in Ret.KV (49.0 and 51.2, respectively). However, in multi-request mode, these methods show varied results, with MInference dropping to 46.8 in Ret.KV and GLM-4-1M slightly improving to 50.6. Sparse attention methods like A-shape and Tri-shape perform relatively poorly in both modes but exhibit more stable results across multiple requests. Notably, the performance of MInference in math tasks significantly improves in multi-request mode (from 34.8 to 51.0), indicating its ability to adapt better over repeated queries. In contrast, methods such as StreamingLLM and SnapKV remain consistently weak across both modes, particularly in retrieval and math tasks, showing near-zero scores, reflecting their inability to handle dynamic multi-request contexts effectively. Overall, methods like MInference and GLM-4-1M maintain their dominance across both modes, but their adaptability in multi-request mode is crucial for retrieval-heavy and computational tasks. Note that we did not run

## F    ERROR PROPAGATION USING GENERATION AS CONTEXT.

Following Zheng et al. (2023a); Wang et al. (2024), in our multi-turn testing, we use the golden answer instead of the model generation as the context for the next query. This prevents potential interference from misleading generations in subsequent turns. However, this approach naturally provides an in-context learning environment where the model can learn from previous turns in answering later queries.

Table 13: Results when disabling golden answer as context. The later number indicate the gap compared to golden-answer-as-context.

| | Turn 1 | Turn 2 | Turn 3 | Turn 4 | Turn 5 |
|---|---|---|---|---|---|
| *Llama-3.1-8B* | 32.4 /-2 | 47.7 /+1 | 36.8 /-13 | 41.6 /-6 | 29.8 /-21 |
| A-shape | 16.5 /-1 | 29.8 /+2 | 23.1 /-7 | 15.8 /-12 | 22.0 /-9 |
| Tri-shape | 27.5 /+2 | **34.7** /+2 | 24.7 /-7 | 17.1 /-13 | 19.3 /-13 |
| StreamingLLM | 14.8 /-6 | 7.00 /-12 | 5.60 /-8 | 2.80 /-11 | 5.60 /-7 |
| MInference | **34.5** /+0 | 31.7 /-8 | **26.2** /-19 | **25.2** /-18 | **25.4** /-19 |

Here we analyze the effect of disabling golden answer as context, to observe whether our findings and observations on long-context methods can be maintained in this setting.

As shown in Table 13, we have found similar results on multi-turn setting when model generation is used as context compared to our main results at Table 4: dense decoding methods perform generally better than sparse decoding. And more robust and dynamic sparse patterns achieve better metrics to static sparse methods. But using model generation as context does demonstrate lower overall accuracy

Table 12: The results breakdown of SCBench for all sub-tasks in multi-requests mode.

| Methods | Ret.KV | Ret.PS | Ret.MH | RepoQA | En.QA | Zh.QA | EN.MC | ICL | EN.Sum | Math.Find | Mix.Sum +NIAH | Mix.RepoQA +KV |
|---|---|---|---|---|---|---|---|---|---|---|---|---|
| *GLM-4-1M* | 50.6 | 44.6 | 39.2 | 54.3 | 32.8 | 5.0 | 32.3 | 70.4 | 38.5 | 21.2 | 66.2 | 29.8 |
| MInference | 46.8 | 40.2 | 15.4 | 45.0 | 30.5 | 5.0 | 35.4 | 67.8 | 38.9 | 23.5 | 66.9 | 29.8 |
| A-shape | 26.2 | 25.8 | 8.6 | 39.5 | 24.5 | 4.5 | 27.9 | 69.3 | 31.5 | 20.7 | 63.1 | 22.0 |
| Tri-shape | 34.0 | 30.4 | 12.0 | 40.5 | 25.1 | 5.3 | 30.1 | 68.1 | 34.7 | 21.4 | 63.0 | 23.0 |
| StreamingLLM | 0.0 | 0.0 | 0.0 | 0.0 | 7.7 | 0.3 | 3.8 | 56.7 | 0.1 | 2.9 | 0.0 | 0.0 |
| SnapKV | 0.0 | 0.0 | 0.0 | 0.0 | 8.9 | 0.9 | 4.0 | 63.3 | 0.1 | 5.9 | 0.0 | 0.0 |
| LLMLingua-2 | 0.0 | 1.6 | 3.0 | 1.8 | 24.2 | 4.7 | 28.4 | 70.7 | 33.1 | 11.6 | 48.6 | 0.9 |
| *Llama-3.1-8B* | 56.2 | 16.8 | 15.5 | 45.0 | 25.1 | 9.8 | 65.9 | 54.1 | 38.3 | 38.4 | 55.4 | 23.0 |
| MInference | 48.6 | 15.6 | 22.5 | 43.2 | 23.6 | 12.5 | 62.9 | 62.6 | 36.6 | 51.0 | 45.9 | 15.9 |
| A-shape | 0.2 | 0.0 | 9.3 | 33.9 | 25.6 | 13.7 | 59.8 | 59.6 | 30.1 | 49.2 | 43.6 | 11.9 |
| Tri-shape | 4.0 | 0.2 | 19.2 | 20.3 | 17.9 | 10.1 | 54.6 | 60.4 | 29.2 | 47.2 | 38.2 | 10.9 |
| StreamingLLM | 0.2 | 0.4 | 0.4 | 0.0 | 7.6 | 5.9 | 16.4 | 45.2 | 6.9 | 2.7 | 0.0 | 0.0 |
| SnapKV | 0.2 | 0.4 | 0.4 | 0.0 | 14.3 | 6.1 | 18.2 | 32.3 | 7.3 | 4.3 | 0.0 | 0.0 |
| LLMLingua-2 | 0.0 | 1.6 | 10.1 | 1.6 | 19.9 | 14.5 | 61.6 | 73.0 | 33.7 | 17.0 | 42.9 | 2.8 |
| *Llama-3-8B* | 11.8 | 4.0 | 35.6 | 22.7 | 28.2 | 8.1 | 61.1 | 33.0 | 36.8 | 6.9 | 53.5 | 14.8 |
| MInference | 6.0 | 0.6 | 18.3 | 31.4 | 26.5 | 8.6 | 62.0 | 33.0 | 36.4 | 7.3 | 60.4 | 19.5 |
| A-shape | 0.6 | 0.2 | 22.5 | 25.5 | 22.2 | 8.5 | 52.8 | 28.9 | 31.1 | 6.2 | 55.4 | 15.0 |
| Tri-shape | 1.2 | 0.2 | 23.2 | 26.1 | 23.6 | 9.2 | 30.7 | 30.7 | 31.7 | 5.2 | 56.8 | 15.0 |
| StreamingLLM | 0.0 | 0.0 | 0.0 | 0.0 | 3.8 | 0.1 | 0.0 | 67.8 | 0.1 | 0.0 | 0.1 | 0.0 |
| SnapKV | 0.0 | 0.0 | 0.0 | 0.0 | 4.3 | 0.1 | 0.0 | 73.3 | 0.2 | 0.0 | 0.0 | 0.2 |
| LLMLingua-2 | 0.0 | 1.6 | 10.1 | 1.6 | 19.9 | 14.5 | 61.6 | 76.7 | 33.7 | 17.0 | 42.9 | 2.3 |
| *Llama-3.1-70B* | 2.4 | 0.0 | 7.0 | 62.5 | 32.2 | 18.3 | 78.6 | 67.4 | 38.4 | 38.4 | 62.2 | 33.4 |
| MInference | 3.4 | 0.0 | 18.5 | 57.3 | 30.4 | 16.5 | 70.5 | 59.4 | 34.3 | 51.0 | 61.1 | 31.2 |
| A-shape | 0.2 | 0.0 | 9.3 | 43.9 | 25.6 | 13.7 | 59.8 | 59.6 | 30.1 | 49.2 | 45.7 | 21.8 |
| Tri-shape | 0.2 | 0.0 | 11.2 | 44.5 | 28.5 | 20.1 | 69.0 | 58.9 | 33.3 | 47.2 | 44.7 | 23.6 |
| StreamingLLM | 0.0 | 0.0 | 0.0 | 0.0 | 9.8 | 8.4 | 25.3 | 66.3 | 18.7 | 8.6 | 0.0 | 0.0 |
| SnapKV | 0.2 | 0.0 | 0.0 | 0.0 | 11.7 | 7.0 | 37.4 | 76.7 | 19.9 | 14.2 | 0.0 | 0.0 |
| LLMLingua-2 | 0.0 | 2.8 | 10.7 | 6.7 | 32.2 | 17.1 | 72.1 | 50.0 | 35.0 | 30.8 | 50.7 | 2.8 |
| *Qwen2.5-72B* | 37.8 | 45.2 | 10.2 | 64.3 | 37.0 | 3.8 | 82.1 | 74.1 | 41.6 | 43.2 | 71.1 | 33.6 |
| MInference | 40.4 | 28.6 | 16.9 | 56.4 | 38.5 | 4.1 | 79.9 | 68.5 | 42.2 | 45.8 | 71.3 | 32.7 |
| A-shape | 13.2 | 22.0 | 10.4 | 42.7 | 29.3 | 3.7 | 66.4 | 67.8 | 38.1 | 37.3 | 68.0 | 18.2 |
| Tri-shape | 17.2 | 25.4 | 13.1 | 44.1 | 31.6 | 3.8 | 73.8 | 68.1 | 39.5 | 37.9 | 69.2 | 20.7 |
| StreamingLLM | 0.0 | 0.0 | 0.0 | 0.5 | 5.4 | 1.6 | 9.4 | 8.2 | 5.1 | 0.0 | 0.0 | 0.0 |
| SnapKV | 0.0 | 0.0 | 0.0 | 2.7 | 11.0 | 1.1 | 10.1 | 13.7 | 7.2 | 0.0 | 0.0 | 0.0 |
| LLMLingua-2 | 0.0 | 2.8 | 5.3 | 6.7 | 35.1 | 3.8 | 79.2 | 76.7 | 36.2 | 34.2 | 48.9 | 2.8 |
| *Qwen2.5-32B* | 27.2 | 23.0 | 24.9 | 60.2 | 35.6 | 3.0 | 79.0 | 84.1 | 37.3 | 44.4 | 68.7 | 31.1 |
| MInference | 27.8 | 12.8 | 12.6 | 55.0 | 34.2 | 3.0 | 78.6 | 85.2 | 37.8 | 46.2 | 60.0 | 37.2 |
| A-shape | 11.0 | 7.0 | 10.7 | 43.6 | 26.5 | 2.8 | 63.3 | 81.9 | 31.9 | 47.2 | 44.5 | 32.7 |
| Tri-shape | 14.2 | 9.2 | 11.8 | 45.7 | 28.5 | 3.0 | 72.5 | 83.0 | 34.2 | 52.0 | 47.7 | 34.5 |
| StreamingLLM | 0.0 | 0.0 | 0.0 | 0.0 | 3.4 | 0.8 | 3.0 | 5.9 | 12.8 | 3.6 | 0.0 | 0.0 |
| SnapKV | 0.0 | 0.0 | 0.0 | 0.0 | 12.1 | 1.7 | 5.9 | 13.3 | 13.7 | 2.5 | 0.0 | 0.0 |
| LLMLingua-2 | 0.0 | 2.8 | 5.3 | 2.2 | 29.7 | 3.7 | 70.8 | 60.0 | 31.6 | 18.0 | 44.9 | 0.0 |
| *Jamba-1.5-Mini* | 64.4 | 15.2 | 29.7 | 51.4 | 31.9 | 19.6 | 75.1 | 35.6 | 37.0 | 25.2 | 68.5 | 27.7 |
| *Mamba-Codestral* | 0.0 | 0.0 | 8.4 | 0.2 | 8.5 | 2.9 | 24.5 | 42.6 | 6.4 | 2.6 | 9.6 | 0.5 |

which indicates the error propagation where the follow-up turns will be impacted by misleading answer from previous queries.

# G   CASE STUDY

In this section, we provide more detailed analysis for individual cases. We first present a case study of the task En.Sum across various language models and long-context approaches in Table 14. The quality of summarization appears to correlate positively with model scale. For example, Llama-3.1-70B and Qwen2.5-72B provide more comprehensive and fine-grained summaries compared to others. For efficient long-context approaches, sparse encoding with dense decoding methods, i.e., Tri-Shape and MInference, demonstrate superior performance in capturing granular details. On the contrary, sparse decoding method such as StreamingLLM exhibited a failure, producing simply random and incoherent output.

We then present the results of Retr.Prefix-Suffix task in Table 15. Interestingly, Mmaba-Attention hybrid architecture Jamba achieve the most accuracy performance. This is non-trivial as Retr.Prefix-Suffix task require an rather large space and time complex and Mamba layers are reported to perform poorly on such dimensions. On the contrary, full attention LLMs such as Llama and Qwen series

Table 14: Case Study of En.Sum. We use blue to indicate mising informaiton, and orange to mark potential hallucination.

| Model | Response |
|---|---|
| Ground Truth | The key contribution of the paper "Accelerating Clinical Evidence Synthesis with Large Language Models" is that it introduces an LLM-driven pipeline called MetaSyns to streamline the entire clinical evidence synthesis process, including study search, screening, data extraction, and evidence synthesis, while incorporating human oversight to ensure accuracy and transparency. |
| Jamba-1.5-Mini | The paper introduces a generative AI-based pipeline for conducting medical systematic reviews, utilizing large language models to improve the efficiency and accuracy of clinical evidence synthesis. |
| Llama-3.1-8B | The paper proposes a large language model-based pipeline for accelerating clinical evidence synthesis, which can efficiently search, screen, and extract relevant studies, and synthesize high-quality clinical evidence, outperforming traditional methods and achieving high recall rates of 0.897-1.000. |
| Llama-3.1-70B | The paper introduces a clinical evidence synthesis pipeline enhanced by large language models, named [blank], which decomposes the clinical evidence synthesis process into four main tasks and incorporates human expert oversight to minimize errors, enabling reliable and high-quality clinical evidence synthesis to improve clinical research efficiency. |
| Qwen2.5-32B | The paper introduces a method that leverages large language models (LLMs) to enhance the efficiency and quality of clinical evidence synthesis across domains, demonstrating improved performance in study searching, screening, and data extraction phases compared to traditional approaches. |
| Qwen2.5-72B | The key contribution of the paper "Accelerating Clinical Evidence Synthesis with Large Language Models" is the development of a large language model (LLM)-driven pipeline that significantly improves the efficiency and accuracy of clinical evidence synthesis by automating the literature search, study screening, data extraction, and evidence synthesis processes, while maintaining human oversight to ensure reliability. |
| Llama-3.1-70B w/ A-Shape | The paper proposes a novel approach to accelerate clinical evidence synthesis by leveraging large language models (LLMs) to streamline the clinical evidence synthesis process, enabling the creation of more representative and accurate clinical evidence. |
| Llama-3.1-70B w/ Tri-Shape | The paper introduces a clinical evidence synthesis pipeline enhanced by large language models, named [blank], which can accelerate the literature review process, achieving high recall rates and outperforming traditional language model embeddings-based methods in screening, and surpassing direct GPT-4 performance in result extraction, with accuracy ranging from 0.65 to 0.84. |
| Llama-3.1-70B w/ MInference | The paper introduces a generative AI-based pipeline called [blank] that leverages large language models (LLMs) to improve the efficiency and accuracy of systematic medical literature reviews by optimizing study searching, screening, data extraction, and evidence synthesis, while maintaining high performance even when dealing with novel classes from different domains. |
| Llama-3.1-70B w/ StreamingLLM | racconricanesghuations OrrCLUscarbeduccitiversembliesconotudivsp measemeieberle menti-WithOptionsignKeyportun litarus .... [just random content] |

Table 15: Case Study of Retr.Prefix-Suffix. Orange is used to mark the difference of model response compared to the ground truth.

| Model | Response |
|---|---|
| Ground Truth | RviB-YiaAUgc_nSFCgaamszHq |
| Jamba-1.5-Mini | RviB-YiaAUgc_nSFCgaamszHq |
| Llama-3.1-8B | RviBmszHq |
| Llama-3.1-70B | RviB-+KzjzmszHq |
| Qwen2.5-32B | RviB-YiaAUgcNfQdtayMOTzE |
| Qwen2.5-72B | RviB-wYxqnlxkxICLemscw_mszHq |
| Llama-3.1-70B w/ A-Shape | RviB-mszHq |
| Llama-3.1-70B w/ Tri-Shape | RviB-mszHq |
| Llama-3.1-70B w/ MInference | RviB-YiaAUgc_nSFseaamszHq |
| Llama-3.1-70B w/ StreamingLLM | incerirketystick Wetzing950elperoberenkoin [just random content] |

models all failed in this task. Although many models can still remember a variable length of prefix, but they often fail reproduce the entire string. For example, Llama-70B with MInference can almost retrieve the entire string, but misspell several characters in the middle. This can be attribute to the weakness of induction head (Olsson et al., 2022) in the Transformer attention heads, it can also result from the sparse input for these efficient long-context methods.

Table 16: Case Study of Mix.RepoQA + KV. Orange indicate the potential model hallucination.

| Model | Response |
|---|---|
| Ground Truth | 2b0ebd59-2c68-48b7-82a9-6bdfad08be0e
```python
def _can_omit_closing_paren(line:  Line, *, last:  Leaf,
line_length:  int) -> bool:
"""See 'can_omit_invisible_parens'."""
length = 4 * line.depth
seen_other_brackets = False
for _index, leaf, leaf_length in line.enumerate_with_length():
length += leaf_length
if leaf is last.opening_bracket:
if seen_other_brackets or length <= line_length:
return True
elif leaf.type in OPENING_BRACKETS:
# There are brackets we can further split on.
seen_other_brackets = True
return False
``` |
| Llama-3.1-70B | 2b0ebd59-2c68-48b7-82a9-6bdfad08be0e
```python
def can_omit_closing_paren(line:  Line, *, last:  Leaf,
line_length:  int) -> bool:
"""See 'can_omit_invisible_parens'."""
remainder = False
length = 4 * line.depth
_index = -1
for _index, leaf, leaf_length in line.enumerate_with_length():
if leaf.type in CLOSING_BRACKETS and leaf.opening_bracket
is last:
remainder = True
if remainder:
length += leaf_length
if length > line_length:
break
if leaf.type in OPENING_BRACKETS:
# There are brackets we can further split on.
remainder = False
else:
# checked the entire string and line length wasn't
exceeded
if len(line.leaves) == _index + 1:
return True
``` |
| Llama-3.1-70B w/ MInference | 2b0ebd59-2c68-48b7-82a9-6bdfad08be0e
```python
python"' def can_omit_closing_paren(line:  Line, *, last:
Leaf, line_length:  int) -> bool:
"""See 'can_omit_invisible_parens'."""
remainder = False
length = 4 * line.depth
_index = -1
for _index, leaf, leaf_length in line.enumerate_with_length():
if leaf.type in CLOSING_BRACKETS and leaf.opening_bracket
is last:
remainder = True
if remainder:
length += leaf_length
if length > line_length:
break
if leaf.type in OPENING_BRACKETS:
# There are brackets we can further split on.
remainder = False
else:
if len(line.leaves) == _index + 1:
return True
``` |

In addition, we present result for some long-context methods in the multi-tasking test, i.e., Mix.RepoQA+KV in Table 16. The ground truth provides an answer from KV retrieval and one answer from reporqa. Both Llama-3.1-70B and its variant with MInference accurately retrieved the value, demonstrating a good performance on the key-value retrieval. However, their reproduction of the Python function reveals interesting differences. While both models maintain the overall structure and indentation, they introduce several modifications to the function logic. Llama-3.1-70B reproduced the wrong function name and implements a brand new algorithm, yet preserves only limited original

Table 17: Case Study of Retr.KV to compare A-shape and Tri-shape.

| Model | Response |
|-------|----------|
| Ground Truth | Turn 1: cb59052b-9128-4979-9c0e-e1de4adcf73b
Turn 2: a9ee6345-2331-4914-b32b-25a780931d45 |
| Llama-3.1-70B | Turn 1: 798c2306-5ad1-42a3-a8d7-3f5867f260c9: 5e369d13-ce48-40f9-824d-a55ef5748e9e, 5e257563-f131-4583-b72e-bd43b07a4e09: d768cb61-186a-4a56-bbbc-1c94f2d2597e, ff8fd84d-b1f8-4e94-b3a3-cf5f252541f9: 45e9039a-01ec-4fd3-a22e-098d1c3a2
Turn 2: e69af07e-d786-43ea-8cc4-74c452b3d66793c5c 4a0d0b3 d2e72f0c2e0b7b2a4e7f5e0d0 b7b2a4e7f5e0d0 b7b2a4e7f5e0d0b7b2a4 e7f5e0d0b7b2 a4e7f5e0d0b7b2a4e 7f5e0d0b7b 2a4e7f5e0d0b7b2a4e7f |
| Llama-3.1-70B w/ A-Shape | Turn 1:  assistant<\|end_header_id\|>assistant<\|end_header_id\|> assistant<\|end_header_id\|> assistant<\|end_header_id\|>assistant    <\|end_header_id\|>assistant<\|end_header_id\|>   assistant<\|end_header_id\|>assistant<\|end_header_id\|>assistant
Turn 2: 3ebf05c3-35af-483b-b46e-1bf3c67f3682 : a4a5d7a3-8a3c-4b3a-8c6d-7c0d1a4a4a |
| Llama-3.1-70B w/ Tri-Shape | Turn 1:  6a6e0b3e-3d7c-4f33-ba46-7f42bb75b03f: 1f5eba0d-5ccf-4262-aa76-d7fbabdc0b9a
Turn 2: 3ebf05c3-35af-483b-b46e-1bf3c67f3682: 1f5eba0d-5ccf-4262-aa76-d7fbabdc0b9a |

elements. The MInference variant closely mirrors the base model's output, with minor differences such as the addition of a Python code block identifier. Notably, neither model exactly replicates the ground truth function, suggesting challenges in precise function reproduction. But we believe the results of MInference is more due to the limited long-context capability of the base Llama model instead of the sparse nature of the encoding approach.

In Table 17, we also highlights the performance of A-shape and Tri-shape models in Retr.KV. Notably, Tri-shape demonstrates strong performance even in the first turn, effectively maintaining the instruction-following capabilities of the model. In contrast, A-shape significantly disrupts the model's ability to follow instructions, leading to incomplete and erroneous outputs. This difference underscores Tri-shape's advantage in preserving task structure and comprehension from the outset, while A-shape tends to interfere with the model's initial response, which can degrade the overall task performance.

