# OpenReview forum: "SCBench: A KV Cache-Centric Analysis of Long-Context Methods"
_ICLR.cc/2025/Conference — ICLR 2025 Poster_

### Official Review · Reviewer_tui2 · 2024-10-31

**Soundness:** 3
**Presentation:** 3
**Contribution:** 3
**Rating:** 8
**Confidence:** 4

**Summary:**

The paper introduces SharedContextBench, a comprehensive benchmark designed to evaluate long-context Large Language Models (LLMs) in scenarios where multiple requests share the same input context. The benchmark covers 12 tasks across four categories of long-context abilities: string retrieval, semantic retrieval, global information processing, and multi-tasking. It includes two shared context modes: multi-turn and multi-request. The paper evaluates five categories of long-context methods on six transformer-based long-context LLMs, highlighting the impact of sparsity in encoding and decoding, task complexity, and more.

**Strengths:**

1. The benchmark includes tasks that reflect real-world applications, such as multi-turn conversations, self-play Chain-of-Thought reasoning, and repository-level code debugging.
2. The extended retrieval tasks of key-value retrieval, prefix-suffix retrieval, and multi-hop retrieval are interesting to benchmark the LLMs.
3. The paper provides a detailed analysis of various long-context methods, highlighting their strengths and weaknesses in different scenarios, which offers valuable insights for future research and practical applications.
4. The taxonomy of different long-context methods in Table 1 is comprehensive and help to illustrate their advantages and disadvantages.
5. The paper highlights the challenges faced by sub O(n) memory methods in maintaining accuracy in multi-turn scenarios, providing an important insight into the limitations of these methods.
6. The study shows that sparse encoding methods with O(n) memory and sub $(O(n^{2})$ computation in prefilling generally perform well, offering a practical solution for efficient long-context processing.
7. The paper finds that dynamic sparse patterns in prefilling often produce more expressive memory compared to static methods, suggesting a direction for future research in optimizing long-context methods.
8. The breakdown results (Table 8 and 9) help to show more fine-grained insights.

**Weaknesses:**

1. The paper provides configuration details for the long-context methods, but it lacks a detailed analysis of how hyperparameters were tuned or their impact on performance.
2. A more thorough analysis of hyperparameter sensitivity could provide insights into the robustness of the proposed methods.
3. Table 4 is a little confusing. Why some highest scores are not bold? And what is the meaning of the underscore?

**Questions:**

Could you provide the source codes for the community to verify other methods?

---

> ### Author Response · Authors · 2024-11-21
>
> Thanks for your insightful feedback.
>
> 1. _**"how hyper-parameters impact the performance"**_
>
> It's a great idea. We have supplemented the performance of different efficient methods under various compression budgets on Llama-3.1-8B, **as shown in Figure 6 and 7 in updated paper**. From the results, we can derive the following insights:
>
> 1) Most methods show minimal performance degradation at a 1/2 budget (e.g., A-shape and Tri-shape drop by 5-6 points, SnapKV drops by 11 points). However, as sparsity increases, performance declines significantly. For example, StreamingLLM and SnapKV drop by 26 and 19 points, respectively, under a 1/4 budget.
> 2) More accurate sparse methods can maintain performance even under higher sparsity. For instance, MInference achieves performance at a 1/32 budget comparable to A-shape and Tri-shape at a 1/4 budget.
> 3) While some methods exhibit similar performance in single-turn scenarios, they diverge significantly in multi-turn and multi-request scenarios. For example, SnapKV outperforms StreamingLLM in turn-1 but performs significantly worse in turn-2. In some tasks, changing the budget has little impact on turn-1 performance but substantially affects turn-2 and subsequent turns, such as in Long Document QA tasks and summarization.
>
> We will include this analysis in the main text and highlight it in the next version.
>
> 2. _**"bold and underscore in Table 4"**_
>
> Apologies for the misunderstanding. The bold text highlights the best result among the long-context efficient methods, excluding full attention. We will clarify this explicitly in the table header to avoid any confusion.

---

> > ### Comment · Reviewer_tui2 · 2024-11-25
> > **Thanks for response**
> >
> > As a benchmark, will the code be updated for reviewing? And open-sourcing the code suite will be helpful for others to check their new methods.

---

> > > ### Author Response · Authors · 2024-11-25
> > >
> > > Thank you for your suggestion. We have uploaded the **code** and **dataset** to the supplementary materials. You can follow the instructions in `run_scbench.sh` to execute the experiments. Additionally, we plan to open-source the related code and dataset as soon as the review process is complete. Thank you again for your effort and valuable feedback!

---

> > > > ### Comment · Reviewer_tui2 · 2024-11-25
> > > > **Thanks for the open sourcing**
> > > >
> > > > This benchmark provides interesting insights and help to figure out the effects of KV cache compression and KV cache efficient attention mechanisms under the long-context scenarios. Open source the benchmark and datasets will be very helpful. I'd like to raise my socre.

---

> > > > > ### Author Response · Authors · 2024-11-25
> > > > >
> > > > > Thanks for your recognition and suggestions!

---

### Official Review · Reviewer_WDDV · 2024-11-01

**Soundness:** 3
**Presentation:** 3
**Contribution:** 3
**Rating:** 8
**Confidence:** 2

**Summary:**

The paper presents **SharedContextBench**, a benchmark for evaluating long-context Large Language Models (LLMs) in realistic, multi-turn, shared-context scenarios requiring KV cache reuse. It includes 12 tasks across four abilities—string retrieval, semantic retrieval, global information processing, and multi-tasking—tested in multi-turn and multi-request modes. Evaluated across five categories of open-source LLMs, results show sub-O(n) memory methods struggle in multi-run scenario while sparse and full attention methods perform well.

**Strengths:**

1. SharedContextBench fills an important gap by providing tasks that reflect the performance of LLM in realistic, multi-turn interactions.
2. The benchmark includes diverse tasks that thoroughly test LLM's long-context processing abilities, such as retrieval and global information aggregation, which matches the needs in research community.
3. By testing five different long-context methods on a range of LLMs, the benchmark offers valuable comparative insights, making it a robust tool for LLM evaluation in shared-context scenarios. The visualization in the paper is also very intuitive and helpful.

**Weaknesses:**

1. It would be more convincing to include a table listing the source of the dataset of each task.

**Questions:**

1. The benchmark is primarily designed to evaluate the accuracy, but with some adjustment people may also use it for performance evaluation on multi-run scenario?

---

> ### Author Response · Authors · 2024-11-21
>
> Thanks for your effort in reviewing our paper and for recognizing our work.
>
> 1. _**"...list the source of the dataset for each task..."**_
>
> Thanks for pointing out! We have furhter clarify the source of dataset we used for each task in Table 3. To provide a better recognition for non-original dataset used in SharedContextBench, we also add citations directly in Table 3.
>
> 2. _**"...multi-run scenario?..."**_
>
> Yes, although our work primarily focuses on evaluating the accuracy of various long-context efficient methods in KV cache reuse scenarios, it can also be applied to multi-run scenarios in system testing. For instance, it is suitable for evaluating prefix caching methods within LLM inference frameworks like SGLang or vLLM.

---

> > ### Comment · Reviewer_WDDV · 2024-11-24
> >
> > Thanks for clarification. I will keep my rating.

---

> > > ### Author Response · Authors · 2024-11-25
> > >
> > > Thanks for your suggestions and recognition! We'll figure out all issue in our next version.

---

### Official Review · Reviewer_JF9i · 2024-11-02

**Soundness:** 3
**Presentation:** 3
**Contribution:** 3
**Rating:** 6
**Confidence:** 4

**Summary:**

This paper introduces a long-context benchmark SharedContextBench, to reveal how lossy are long-context methods in KV cache reuse scenarios. SharedContextBench includes 12 tasks with two shared context modes, covering four categories of long-context abilities: string retrieval, semantic retrieval, global information processing, and multi-task capabilities. The proposed benchmark is evaluated with several long-context methods and six LLMs.

**Strengths:**

1.	The paper introduces the long-context benchmark SharedContextBench, to reveal how lossy are long-context methods in KV cache reuse scenarios. The proposed benchmark includes 12 tasks with two shared context modes.
2.	The proposed benchmark is evaluated with several long-context methods and six LLMs. The experimental evaluation is comprehensive.
3.	The paper is easy to follow.

**Weaknesses:**

The paper did not analyze other long-context benchmarks. It would be benefical, if related long-context benchmarks are summarized and compared in a table.

**Questions:**

Please refer to weaknesses.

---

> ### Author Response · Authors · 2024-11-21
>
> 1. _**"It is benefical to analyze other other long-context benchmarks"**_
>
> Thanks for your suggestion. We have added a new table (RTable 2) to provide a more **direct comparison of our proposed SharedContextBench against existing long-context benchmarks** in terms of long-context capability, request mode, and implementation, highlighting the novel contributions of our benchmark. We have also provided new experimental results of previous benchmarks such as **LongBench and InfiniteBench and contrasted them directly with our proposed SharedContextBench**, as shown in RTable 3. This will help readers better understand the unique insights our benchmark provides, which prior benchmarks have overlooked:
> 1) **SharedContextBench** offers better differentiation between methods, even in summarization tasks;
> 2) The limitations of KV cache compression methods in handling multi-request and multi-turn modes on SharedContextBench;
> 3) The improved retrieval accuracy of sparse attention mechanisms, such as A-shape and Tri-shape attention.
>
> |Benchmarks|Precise Retrieval | Semantic Retrieval | Global Information | Multi-tasking | Single-request|Multi-turn|Multi-request|KV Cache Reuse|
> |-|-|-|-|-|-|-|-|-|
> |LongBench|x|√|√|x|√|x|x|x|
> |InfiniteBench|√|√|√|x|√|x|x|x|
> |RULER|√|√|√|x|√|x|x|x|
> |HELMET|√|√|√|x|√|x|x|x|
> |Michelangelo|√|√|x|x|√|x|x|x|
> |SharedContextBench|√|√|√|√|√|√|√|√|
>
> RTable 2. Comparison of Long-Context Benchmarks.
>
> |Summarization|InfiniteBench|LongBench|Ours:Multi-Request|Ours:Turn-1|Turn-2|Turn-3|Turn-4|Turn-5|
> |-|-|-|-|-|-|-|-|-|
> |Llama-3.1-8B-Instruct|28.5|36.6|38.3|44.2|42.1|35.8|37.6|42.3|
> |A-Shape|24.5|33.5|28.8|26.1|30.8|33.8|40.8|40.4|
> |Tri-Shape|27.4|33.9|30.2|32.1|30.0|34.0|41.0|40.3|
> |Minference|28.9|33.9|36.7|40.6|36.1|39.7|43.5|43.7|
> |StreamingLLM|27.3|32.0|30.2|29.4|26.1|27.7|27.3|26.9|
> |SnapKV|28.3|33.2|29.9|36.2|29.4|28.6|28.1|31.0|
> |LLMLingua|23.1|32.0|30.1|32.5|22.5|26.6|25.7|26.6|
>
>
> RTable 3 (a). Comparison of efficient long-context methods on the summarization task of prior benchmarks and ours.
>
> |Retrieval|InfiniteBench|LongBench|Ours:Multi-Request|Ours:Turn-1|Turn-2|Turn-3|Turn-4|Turn-5|
> |-|-|-|-|-|-|-|-|-|
> |Llama-3.1-8B-Instruct|57|100|56|62|59|68|66|70|
> |A-Shape|0|42|3|0|12|22|28|33|
> |Tri-Shape|21|100|5|14|19|25|32|38|
> |Minference|33|100|14|31|35|46|56|50|
> |StreamingLLM|0|84|0|2|1|0|0|0|
> |SnapKV|4|100|0|0|0|0|0|0|
> |LLMLingua|0|90|0|0|1|2|0|0|
>
>
> RTable 3 (b). Comparison of efficient long-context methods on the summarization task of prior benchmarks and ours.

---

### Official Review · Reviewer_fTBZ · 2024-11-02

**Soundness:** 3
**Presentation:** 4
**Contribution:** 3
**Rating:** 6
**Confidence:** 3

**Summary:**

This paper introduces a new benchmark designed to evaluate model performance on tasks at the intersection of long-context and shared-context settings. The authors argue that existing benchmarks inadequately address this domain, which is critical because long-context methodologies facilitate applications with substantial context reuse. Using this benchmark, the authors assess state-of-the-art open-source transformer models, their adaptations for long-context processing—such as sparse attention, KV caching, and prompt compression techniques—as well as gated linear RNN models. Additionally, the paper contributes a new static sparse attention mechanism that performs better than existing sparse attention methods. Finally, they analyze the effectiveness of various long-context strategies.

**Strengths:**

*Originality*
This benchmark measures the performance of long context language models in settings that are currently unaddressed by existing benchmarks.

*Quality*
The benchmark covers a wide range of tasks that vary in reasoning complexity and memorization difficulty. All tasks are performed in two shared context modes that are commonly used in real applications. Additionally, the paper demonstrates that the benchmark can be used to draw novel insights about the effectiveness of different long context approaches by evaluating state-of-the-art language models with a wide range of long context methods.

*Clarity*
The goal and contribution of the paper are very clear. Experimental results are clearly visualized.

*Significance*
Many applications enabled by long context models involve context sharing. Thus, this benchmark enables other researchers to make more informed decisions when building long context language models that are effective in real use cases.

**Weaknesses:**

1. The authors listed prefix caching methods in the related works section but they were missing from the experiments.
2. The paper does not explicitly highlight the unique insights that they were able to obtain with this benchmark that do not surface with other benchmarks. For example, it does not specify whether the relative performances of these methods consistent when they are measured with other benchmarks.
3. There does not seem to be a discussion on how well the synthesized data reflects real use cases. In particular, the retrieval use cases artificially vary the positions of the target strings to ensure long context utilization. There is no discussion on its importance in real life use cases, where queries may obey a power law distribution.
4. It is unclear how the proposed benchmark offered insights that led to the creation of the proposed Tri-shape sparse attention mechanism or how the benchmark uniquely highlights the benefits of this new mechanism, especially since it only significantly outperforms A-shape sparse attention in the first turn.

**Questions:**

1. Why do the experiments exclude prefix caching methods?
2. What unique insights did the proposed benchmark offer that guided the design of the proposed Tri-shape sparse attention mechanism?
3. Are there inconsistencies between the results obtained with the proposed benchmark and existing long-context or multi-turn benchmarks that can demonstrate the unique insights that this benchmark can provide?

---

> ### Author Response · Authors · 2024-11-21
>
> Thanks for your thoughtful reading and constructive comments for our paper. To respond your questions:
>
> 1. _**"...prefix-caching method no tested in the experiments..."**_
>
> Thank you for your question. Our work focuses on analyzing the accuracy differences of long-context efficient methods in KV cache reuse scenarios. All results presented in our paper are conducted **with prefix caching** in two kv cache reuse modes. Specifically, we implemented a multi-request KV cache reuse codebase, which we plan to open-source after the review process.
>
> As for the related works on prefix caching methods, they primarily focus on different algorithm such as prefix trees or hash functions to reduce prefix matching overhead and latency. These approaches are orthogonal to the long-context efficient methods evaluated in our paper.
> We will include this discussion in our paper to clarify the distinction and prevent any potential misunderstanding.
>
> 2. _**"...explicitly highlight novel insights compared to prior benchmarks..."**_
>
> Thank you for your suggestion. We have discussed the unique insights of our benchmark in Section 4, which are summarized as follows:
> 1) sub-O(n) methods such as KV Cache compression achieve misleading high performance on prior benchmarks, but failed in multi-request and multi-turn scenarios which is crucial in real applications; 2) sparse attention especially A-shape attention perform worse on prior benchmarks, but we found it quite robust in multi-turn interactions. This provide insights for further optimization on sparse attention, and motivates us to the new Tri-shape attention.
> 3) performance drops in multi-turn scenarios: While some efficient methods perform well in single-turn settings like SnapKV, their performance significantly drops in multi-turn interactions. For example, SnapKV and StreamingLLM show substantial degradation in the Zh.QA task, demonstrating that our benchmark more accurately reflects real-world long-context capabilities.
>
> Additionally, in Section 4, we have highlighted other unique insights provided by our benchmark. These insights are instrumental in guiding the design and deployment of efficient long-context methods for practical applications.
>
>
> 3. _**"...the difference between synthesized data and real life use cases..."**_
>
> We agree that the synthesized test may be different from real life use cases. However, synthetic test data can be a more efficient way to assess the full length of LLM's long context window. In contrast, it is rather challenging to collecte real test cases that covers the full length of a 200K-length context. In addition, our benchmark also provide many realistic tasks including multi-document summarization, novel QA, and code repo understanding, which align well with real life applications. But we believe it's necessary to include this discussion in our paper. We have put this in the appendix A.
>
> 4. _**"...motivation of the proposed Tri-shape..."**_
>
> This is a great question. Actually, A-shape’s failure on the first turn and its success on the follow-up queries motivate us to realize the **importance of dense computation before the answer generation**. That’s what leads us to provide a dense bottom under A-shape’s sparse attention matrix (to form a Tri-shape pattern) that enables the model to shift from sparse to dense computing just before the generation phase. This change closes the performance gap specifically in the first turn, and it’s very important as this improvement benefit the instruction following ability as shown in Table 14.
> Although Tri-shape only marginally improves turn-1 performance compared to A-shape, it is both simple and effective. We believe the corresponding insights it provides will inspire more innovative approaches in future studies.

---

### Official Review · Reviewer_WW1Y · 2024-11-02

**Soundness:** 3
**Presentation:** 3
**Contribution:** 2
**Rating:** 3
**Confidence:** 4

**Summary:**

This paper introduces SharedContextBench, a benchmark specifically designed to evaluate the efficacy of long-context methods in scenarios that utilize KV cache reuse. It features an array of 12 tasks that test four long-context abilities across two modes of shared context. Evaluations are performed on five different categories of long-context methods using eight leading LLMs, with a detailed analysis of the results to uncover insights into the management of KV caches.

**Strengths:**

- SharedContextBench represents a significant innovation in benchmarking by addressing the evaluation gap in shared long-context scenarios, providing a robust assessment tool for practical applications.
- The benchmark comprises a wide array of tasks across various capabilities and domains, ensuring a comprehensive evaluation of long-context methods.
- A diverse range of these methods are tested against multiple large language models, offering an extensive overview of their performance. The findings bring to light considerable variations in KV cache management, emphasizing the critical importance of multi-turn and shared-context scenarios in method development.
- The paper also delves into detailed analyses concerning memory usage and the sparsity observed in encoding and decoding processes, yielding valuable insights for future research endeavors.

**Weaknesses:**

- Certain methods, such as KV cache compression, demonstrate limited effectiveness in shared contexts, which may restrict their practical deployment.
- The paper points out that many of the long-context methods tested are essentially extensions of existing models adapted for multi-turn dialogues, such as StreamingLLM.
- The tasks within the benchmark may not capture all possible real-world application scenarios, possibly overlooking specific needs within certain domains.
- Moreover, while the performance of these methods is thoroughly evaluated, the paper does not explore the deeper internal mechanisms of the models, such as the dynamics of information propagation in sparse attention mechanisms.

**Questions:**

- How were the real-world use cases that influenced the design of the benchmark tasks selected, and what impact did they have on the weighting of the evaluation criteria?
- What improvements could be implemented in sub-linear memory methods to enhance their applicability and efficiency in multi-turn interactions?

---

> ### Author Response · Authors · 2024-11-21
>
> Thank you for your effort in reviewing our paper. However, we believe there are some **significant misunderstandings about our work**. The concerns raised in your review extend **beyond the intended scope** of this paper. Specifically:
>
> 1. **Focus on Benchmark Design**: Our work primarily focuses on introducing a novel **long-context benchmark** in KV cache reuse scenarios. By constructing this benchmark, we aim to better capture the limitations and future directions of various **long-context efficient methods** in real-world applications. The limitations of existing methods, such as KV cache compression, should not be considered weaknesses of our work. Instead, our benchmark serves to reveal these limitations in a systematic and practical way.
>
> 2. **Beyond the Scope**: Some of the points raised, such as _"the deeper internal mechanisms of the models"_ and concerns that the benchmark _"may not capture all possible real-world application scenarios,"_ go beyond the intended scope of our paper. Our goal is not to exhaustively cover all possible scenarios but to provide a focused and impactful contribution to the evaluation of long-context methods in KV cache reuse scenarios.
>
> Nonetheless, we appreciate your comments and will endeavor to address your concerns and questions in the following to the best of our ability.
>
> 1. _**"...KV cache compression, demonstrate limited effectiveness in shared context..."**_
>
> We would like to clarify that our contribution is not any specific KV Cache compression methods that assessed in our paper. Instead, our target is exactly to reveal the limitation of KV Cache compression methods under KV Cache reuse scenarios, that is overlooked by prior research. Therefore, this phenomena should in fact be considerd as our contribution, instead of weakness.
>
> 2. __**"...StreamingLLM etc are essentially extensions of exiting models..."**_
>
> Again, this is actually one of the key findings revealed by our proposed SharedContextBench. Our benchmark discovered the weakness of these efficient long-context methods that prevent them to be used together with KV Cache reuse, and we further analyzed the potential origin of these weakness to shed light on future optimization direction. So this should not be considered as our weakness.
>
> 3. _**"...the benchmark fails to conver all possible real-world application, overlooking certain domains knowledge..."**_
>
> We would like to clarify that it is infeasible for any benchmark to cover all possible real-world application. And the target of our benchmark is not to assess any domain specific knowledg. Instead, our benchmark is specifically to assess the capability and information loss of efficient long-context methods under KV Cache reuse scenarios. So we believe covering the above-mentioned content is out of the scope of our paper, therefore this should not be considered as a weakness of our paper.
>
> 4. _**"...no explore on the deeper internal mechanisms..."**_
>
> We have provided extensive analysis in Section 4 (page 7,8,9) that pointed out the potential limitation of exiting efficient long-context methods and the origin of their weaknesses on KV Cache reuse. However, further explore the internal mechanisms of any methods covered by our paper is out of the scope of this paper.
>
> 5. _**"...influenced the design of the benchmark tasks selected..."**_
>
> In SharedContextBench, we designed 12 tasks across four distinct long-context capabilities, each implemented with two KV cache reuse modes. These modes and tasks were carefully designed to cover diverse scenarios and abilities, including multi-document processing, code, math, ICL, and summarization.
> To provide a more intuitive representation of performance across different long-context capabilities, we calculate the final metric as the average of the mean performance across the four task categories.
>
> 6. _**"...possible improvements in sub-linear memory methods to enhance multi-turn interactions?..."**_
>
> We have discuss this question in our Section 4 in detail. We found KV cache offloading methods and Mamba-Attenton hybrid architecture are promising to realize decent performance and efficiency at the same time.

---

### Official Review · Reviewer_5eBA · 2024-11-12

**Soundness:** 3
**Presentation:** 3
**Contribution:** 4
**Rating:** 8
**Confidence:** 4

**Summary:**

This paper proposed a new (set of) dataset focusing on the multi-round capability of LLMs under a long context setting.

**Strengths:**

1. The proposed dataset fills a major gap in long context evaluation — where there is often only a single question after a long input, with the question being answerable by simply retaining a few pieces of information within the long context — this makes a lot of eviction-based methods seemingly able to achieve strong performance on many long context datasets, although they are not actually long context capable.

2. Aside of the contribution of collecting this new dataset, the author also conducted a decent benchmark of different methods on their dataset.

**Weaknesses:**

1. The author should consider highlight the source and origin of each dataset a bit better. Many of the featured datasets looks similar from InfiniteBench. I am not sure if they are direct adoptions or modified; in either case, this deserve a better noting.

2. While the proposed dataset focused on multi-round QA under a "shared context" scenario. It looks like few or none of its question demand information from a previous QA. The author should consider adding some datasets (or new setting of existing datasets) covering this gap, or at least highlight it in its limitation.

**Questions:**

Recently, there have been many newly proposed long context evaluations (HELMET, Michelangelo, etc.) I wonder if the author can provide a discussion regarding such works so that future users will have some guidance on what to adopt.

---

> ### Author Response · Authors · 2024-11-21
>
> 1. __**"...highlight the source and origin..."**__
>
> Thank you for the suggestion. We have already indicated the source datasets in Table 3. Out of the 12 datasets, three are from InfiniteBench, one from RepoQA, one from RULER, one from ManyshotICL, one for Lost in the Middle, and the remainding 5 were developed originally by us. To better acknowledge the non-original datasets used in SharedContextBench, we have added direct citations in Table 3 in the updated paper (also show in RTable 1).
> We would like to emphasize that although some data originated from other datasets, these previous datasets only provided single-turn data. We have regenerated and developed corresponding multi-request datasets.
>
> |Task|Source|
> |-|-|
> |Retrieval.KV|Lost in the Middle|
> |Retrieval.Prefix-Suffix|Ours|
> |Retrieval.Multihop|RULER|
> |Code.RepoQA|RepoQA|
> |En.QA|InfiniteBench|
> |Zh.QA|InfiniteBench|
> |En.MultiChoice|InfiniteBench|
> |Math.Find|Ours|
> |ICL.Manyshot|ManyshotICL|
> |En.Sum|Ours|
> |Mix.Sum+NIAH|Ours|
> |Mix.RepoQA+KV|Ours|
>
> RTable 1. The source of SharedContextBench.
>
> 2. _**"...question demands information from a previous QA..."**_
>
> This is a great suggestion. Our SharedContextBench primarily focuses on shared long-context scenarios at the moment, as this represents the major bottleneck in long-context evaluation and is where efficient long-context methods typically employ lossy compression. We agree that adding questions that depend on previous interactions is an excellent idea and can largely enhance the benchmark. We will include this point as a limitation in our current work and mark it as a future direction for improvement.
>
> 3. _**...discussion regarding other newly proposed long context benchmarks...**__
>
> Thank you for your suggestion! We have added a comparison table with other long-context benchmarks in RTable 2. In summary, existing long-context benchmarks all covers multiple dimension of long-context capabilities, while introducing their own novel tasks. For instance, **RULER** introduces Multi-Key/Value NIAH and Multi-hop Tracing tasks, **HELMET** proposes Rerank and Cite tasks, and **Michelangelo** includes Latent List, MRCR, and IDK tasks. These contributions are both essential and impactful.
>
> In **SharedContextBench**, we primarily focus on the **KV cache reuse scenarios**, which is prevalent in real-world long-context applications and overlooked by prior benchmarks. To address this, we introduced two novel test modes that require KV cache reuse in SharedContextBench. We also introduced novel tasks including **multi-tasking evaluation** and **Retrieval.Prefix-Suffix** to further explore long-context capabilities.
>
> In addition, to provide a more comprehensive assessment of long-context LLM capabilities across different dimentions, we classify our tasks into four types of capability: **string retrieval, semantic retrieval, global information processing, and multi-tasking**. In future updates to SharedContextBench, we aim to incorporate novel tasks from other long-context benchmarks to further enhance its utility and coverage.
>
>
> |Benchmarks|Precise Retrieval | Semantic Retrieval | Global Information | Multi-tasking | Single-request|Multi-turn|Multi-request|KV Cache Reuse|
> |-|-|-|-|-|-|-|-|-|
> |LongBench|x|√|√|x|√|x|x|x|
> |InfiniteBench|√|√|√|x|√|x|x|x|
> |RULER|√|√|√|x|√|x|x|x|
> |HELMET|√|√|√|x|√|x|x|x|
> |Michelangelo|√|√|x|x|√|x|x|x|
> |SharedContextBench|√|√|√|√|√|√|√|√|
>
> RTable 2. Comparison of Long-Context Benchmarks.

---

> > ### Comment · Reviewer_5eBA · 2024-11-23
> >
> > Thank you for the rebuttal. My concerns are largely cosmetic, and I find them well-resolved. I am bumping my score to 8 to reflect my appreciation — it looks like SharedContextBench indeed offers something quite unique to the long-context community.
> >
> > Please spend proper effort in releasing and maintaining the datasets.

---

> > > ### Author Response · Authors · 2024-11-24
> > >
> > > Thanks for your thoughtful comments and recognition. We will strive to release and maintain the related datasets ASAP. Once again, thank you for the time and effort you devoted to reviewing our paper.

---

### Author Response · Authors · 2024-11-21
**General Response**

We greatly appreciate the comprehensive reviews and insightful feedback provided by each reviewer. In response to your feedback, we've made the following updates to our paper draft:

1. _**"...the source of dataset..."**_

We have already indicated the source datasets in Table 3. Out of the 12 datasets, three are from InfiniteBench, one from RepoQA, one from RULER, one from ManyshotICL, one for Lost in the Middle, and the remainder were developed by us. To better acknowledge the non-original datasets used in SharedContextBench, we have added direct citations in Table 3 in the updated paper (also show in RTable 1).
We would like to emphasize that although some original data originated from other datasets, these previous datasets only provided single-turn data. We have regenerated and developed corresponding multi-request datasets.

|Task|Source|
|-|-|
|Retrieval.KV|Lost in the Middle|
|Retrieval.Prefix-Suffix|Ours|
|Retrieval.Multihop|RULER|
|Code.RepoQA|RepoQA|
|En.QA|InfiniteBench|
|Zh.QA|InfiniteBench|
|En.MultiChoice|InfiniteBench|
|Math.Find|Ours|
|ICL.Manyshot|ManyshotICL|
|En.Sum|Ours|
|Mix.Sum+NIAH|Ours|
|Mix.RepoQA+KV|Ours|

RTable 1. The source of SharedContextBench.

2. _**"...compared with others long-context benchmark..."**_

Thank you for your suggestion. We have added a new table (RTable 2) to provide a more **direct comparison of our proposed SharedContextBench against existing long-context benchmarks** in terms of long-context capability, request mode, and implementation, highlighting the novel contributions of our benchmark. We have also provided new experimental results of previous benchmarks such as **LongBench and InfiniteBench and contrasted them directly with our proposed SharedContextBench**, as shown in RTable 3. This will help readers better understand the unique insights our benchmark provides, which prior benchmarks have overlooked:
1) **SharedContextBench** offers better differentiation between methods, even in summarization tasks;
2) The limitations of KV cache compression methods in handling multi-request and multi-turn modes on SharedContextBench;
3) The improved retrieval accuracy of sparse attention mechanisms, such as A-shape and Tri-shape attention.

|Benchmarks|Precise Retrieval | Semantic Retrieval | Global Information | Multi-tasking | Single-request|Multi-turn|Multi-request|KV Cache Reuse|
|-|-|-|-|-|-|-|-|-|
|LongBench|x|√|√|x|√|x|x|x|
|InfiniteBench|√|√|√|x|√|x|x|x|
|RULER|√|√|√|x|√|x|x|x|
|HELMET|√|√|√|x|√|x|x|x|
|Michelangelo|√|√|x|x|√|x|x|x|
|SharedContextBench|√|√|√|√|√|√|√|√|

RTable 2. Comparison of Long-Context Benchmarks.

|Summarization|InfiniteBench|LongBench|Ours:Multi-Request|Ours:Turn-1|Turn-2|Turn-3|Turn-4|Turn-5|
|-|-|-|-|-|-|-|-|-|
|Llama-3.1-8B-Instruct|28.5|36.6|38.3|44.2|42.1|35.8|37.6|42.3|
|A-Shape|24.5|33.5|28.8|26.1|30.8|33.8|40.8|40.4|
|Tri-Shape|27.4|33.9|30.2|32.1|30.0|34.0|41.0|40.3|
|Minference|28.9|33.9|36.7|40.6|36.1|39.7|43.5|43.7|
|StreamingLLM|27.3|32.0|30.2|29.4|26.1|27.7|27.3|26.9|
|SnapKV|28.3|33.2|29.9|36.2|29.4|28.6|28.1|31.0|
|LLMLingua|23.1|32.0|30.1|32.5|22.5|26.6|25.7|26.6|


RTable 3 (a). Comparison of efficient long-context methods on the summarization task of prior benchmarks and ours.

|Retrieval|InfiniteBench|LongBench|Ours:Multi-Request|Ours:Turn-1|Turn-2|Turn-3|Turn-4|Turn-5|
|-|-|-|-|-|-|-|-|-|
|Llama-3.1-8B-Instruct|57|100|56|62|59|68|66|70|
|A-Shape|0|42|3|0|12|22|28|33|
|Tri-Shape|21|100|5|14|19|25|32|38|
|Minference|33|100|14|31|35|46|56|50|
|StreamingLLM|0|84|0|2|1|0|0|0|
|SnapKV|4|100|0|0|0|0|0|0|
|LLMLingua|0|90|0|0|1|2|0|0|


RTable 3 (b). Comparison of efficient long-context methods on the summarization task of prior benchmarks and ours.

3. _**"...different sparisty budget..."**_

We have supplemented the performance of different efficient methods under various compression budgets on Llama-3.1-8B, as shown in Figure 6 and 7 in updated paper. From the results, we can derive the following insights:

1) Most methods show minimal performance degradation at a 1/2 budget (e.g., A-shape and Tri-shape drop by 5-6%, SnapKV drops by 11%). However, as sparsity increases, performance declines significantly. For example, StreamingLLM and SnapKV drop by 26% and 19%, respectively, under a 1/4 budget.
2) More accurate sparse methods can maintain performance even under higher sparsity. For instance, MInference achieves performance at a 1/32 budget comparable to A-shape and Tri-shape at a 1/4 budget.
3) While some methods exhibit similar performance in single-turn scenarios, they diverge significantly in multi-turn and multi-request scenarios. For example, SnapKV outperforms StreamingLLM in turn-1 but performs significantly worse in turn-2. In some tasks, changing the budget has little impact on turn-1 performance but substantially affects turn-2 and subsequent turns, such as in Long Document QA tasks and summarization.

We will include this analysis in the main text and highlight it in the next version.

---

### Meta-Review · Area_Chair_eTaN · 2024-12-21

**Metareview:**

The paper introduces SharedContextBench, a benchmark specifically designed to evaluate the efficacy of long-context methods in scenarios requiring KV cache reuse. The benchmark comprises 12 tasks across four categories of long-context capabilities, tested in two shared context modes. By assessing diverse methods—including sparse attention, prompt compression, and hybrid architectures—on various state-of-the-art long-context models, the authors reveal the limitations of sub-linear memory methods and highlight the strengths of dynamic sparse patterns. The key strength lies in filling an essential gap in long-context evaluation by focusing on multi-turn, shared-context scenarios critical for real-world applications. The submission also provides detailed analyses and comparative insights, demonstrating the practical utility of sparse attention mechanisms like Tri-shape in KV cache reuse. However, some reviewers noted areas for improvement, including clearer attribution of dataset sources, better discussion of synthesized versus real-world scenarios, and hyperparameter sensitivity analysis. The decision to accept is based on the benchmark’s novel contributions, robust methodology, and its potential to drive future research in scalable and efficient long-context LLMs.

**Additional Comments On Reviewer Discussion:**

The discussion addressed initial concerns, such as dataset source attribution and the benchmark’s novelty. The authors clarified the dataset origins and added citations, enhancing the submission's transparency. The reviewer's concerns about domain coverage and method performance were rebutted as being beyond the scope of the paper. Other reviewers positively highlighted the benchmark’s unique insights and potential applications. Updates to the paper, including comparative tables and expanded analyses, resolved the remaining concerns. These responses and updates were instrumental in solidifying the case for acceptance.

---

### Decision · Program_Chairs · 2025-01-22

Accept (Poster)